# On the Sample Efficiency of Inverse Dynamics Models for Semi-Supervised Imitation Learning

**Sacha Morin** [1 2 *]    **Moonsub Byeon** [3]    **Alexia Jolicoeur-Martineau** [4]    **Sébastien Lachapelle** [4]

## Abstract

Semi-supervised imitation learning (SSIL) consists in learning a policy from a small dataset of action-labeled trajectories and a much larger dataset of action-free trajectories. Some SSIL methods learn an inverse dynamics model (IDM) to predict the action from the current state and the next state. An IDM can act as a policy when paired with a video model (VM-IDM) or as a label generator to perform behavior cloning on action-free data (IDM labeling). In this work, we first show that VM-IDM and IDM labeling learn the same policy in a limit case, which we call the IDM-based policy. We then argue that the previously observed advantage of IDM-based policies over behavior cloning is due to the superior sample efficiency of IDM learning, which we attribute to two causes: (i) the ground-truth IDM tends to be contained in a lower complexity hypothesis class relative to the expert policy, and (ii) the ground-truth IDM is often less stochastic than the expert policy. We argue these claims based on insights from statistical learning theory and novel experiments, including a study of IDM-based policies using recent architectures for unified video-action prediction (UVA). Motivated by these insights, we finally propose an improved version of the existing LAPO algorithm for latent action policy learning. We experiment on the Procgen, Push-T and LIBERO benchmarks.

## 1. Introduction

*Behavior cloning* (BC) (Pomerleau, 1988) is a capable technique for learning control policies via supervised learning on action-labeled expert trajectories in both state-based and visual domains. Following the successes in natural language processing and computer vision, applying BC to large datasets is seen as a promising avenue for learning general policies in fields such as robotics (O'Neill et al., 2024). However, scaling datasets for BC requires collecting action-labeled expert demonstrations, a particularly onerous process in domains requiring human demonstrations.

There is therefore a strong interest in leveraging data that is already available "in the wild", typically in the form of videos. Videos tend to be goal-directed and depict quasi-expert or expert behaviors that can be leveraged for policy learning. Such data, however, does not include action labels. Learning from abundant action-free data and a comparatively small dataset of action-labeled data is called *semi-supervised imitation learning* (SSIL) (Baker et al., 2022).

A number of high-performing methods for SSIL leverage an *inverse dynamics model* (IDM) to predict actions from the current and next observations. An IDM can be trained on the small, action-labeled dataset and used to synthetically label action-free data for downstream BC (IDM labeling) (Baker et al., 2022). Other methods form a policy by instead coupling the IDM with a *video model* (VM) trained on unlabeled data (VM-IDM) (Du et al., 2023). The IDM also plays an important roles in recent methods for latent action learning (Schmidt & Jiang, 2024).

To effectively leverage action-free data, IDM-based SSIL methods make the assumption that the IDM will generalize better than BC trained on the same amount of labeled data. While the sample efficiency of the IDM has been measured empirically, previous works only hypothesize partial explanations for this phenomenon, ranging from the IDM being non-causal and simpler (Baker et al., 2022) to analogies with the sample-efficiency of model-based reinforcement learning (Torabi et al., 2018).

In this work, we unify IDM-based methods and seek a more complete explanation for their success in SSIL settings. We argue that the sample efficiency of IDM learning stems from

[1]Département d'informatique et de recherche opérationnelle, Université de Montréal [2]Mila – Quebec AI Institute [3]Samsung Electronics, Suwon [4]Samsung AI Lab, Montreal. [*]Work performed during an internship at Samsung AI Lab, Montreal. Correspondence to: Sacha Morin <sacha.morin@umontreal.ca>.

*Proceedings of the 43rd International Conference on Machine Learning*, Seoul, South Korea. PMLR 306, 2026. Copyright 2026 by the author(s).

the reduced complexity and stochasticity of the ground truth IDM compared to the ground truth policy. These factors are environment-specific and offer a valuable framework for understanding when and to what extent IDM-based learning can outperform BC. Experimentally, we explore IDM-based learning across multiple datasets and show how the statistical advantage of IDM learning can be leveraged to improve other IDM-based SSIL methods. Our key contributions are

1. We show that, at optimality, VM-IDM and IDM labeling recover the same policy when the unlabeled dataset is infinite and model capacity is sufficient. We call this common policy the IDM-based policy (Section 3).
2. We argue that IDM learning will be more sample efficient than BC when: (i) the ground-truth IDM is contained in a lower complexity hypothesis class relative to the expert policy, and/or (ii) the ground-truth IDM is less stochastic than the expert policy. We use insights from statistical learning theory and experiments to argue these claims (Section 4). We also hypothesize that at least one of these two situations is likely to occur in practical settings.
3. We present an extensive and novel comparison across the 16 Procgen environments, Push-T and LIBERO, and discuss how IDM properties correlate with the increased performance of IDM-based policies over BC (Section 5).
4. Motivated by the sample-efficiency of IDM learning, we propose an improved version of the LAPO algorithm for latent action policy learning (Schmidt & Jiang, 2024) and demonstrate its superiority on the Procgen benchmark (Section 5.1). We further show how sampling a recent architecture for unified video-action prediction (UVA) as a VM-IDM can improve policy success (Section 5.2).

Code for all our experiments is available here.

## 2. Background & Related Work

### 2.1. Finite-Horizon Markov Decision Processes (MDP)

We consider a Markov decision process (MDP) with finite-horizon $T$, state space $\mathcal{S}$ (e.g. positions or images), initial state distribution $p^1(s^1)$, action space $\mathcal{A}$, stationary transition kernel $p(s' \mid s, a)$ and reward function $r(s, a, s')$ (Sutton & Barto, 2018). Given a stationary policy $\pi(a \mid s)$, let $d_\pi(\tau)$ be the distribution over sequences $\tau := (s^1, a^1, s^2, a^2, \ldots, s^T, a^T, s^{T+1})$ obtained by following policy $\pi$ at each iteration. Let $p_\pi^t(s)$ be the distribution of the $t$th state $s^t$ when following $\pi$. We can then define the *state visitation distribution* as $p_\pi(s) := \frac{1}{T} \sum_{t=1}^T p_\pi^t(s)$. We also define the *transition visitation distribution* as

$$p_\pi(s, a, s') := p_\pi(s)\pi(a \mid s)p(s' \mid s, a). \tag{1}$$

A sample from the above corresponds to (i) sampling a trajectory $\tau$ from $d_\pi$, (ii) sampling with uniform probability a time step $t \in \{1, \ldots, T\}$, and (iii) outputting $(s^t, a^t, s^{t+1})$.

Let $r(\pi) := \mathbb{E}_{d_\pi(\tau)} \sum_{t=1}^T r(s^t, a^t, s^{t+1})$ be the total expected reward achieved by $\pi$.

### 2.2. Behavior Cloning (BC)

In BC, we are given a dataset of state-action pairs $\mathcal{D}_L := \{(s_i, a_i)\}_{i=1}^{N_L}$ sampled from $p_{\pi^*}$ where $\pi^*$ is an expert policy achieving high reward. BC consists in learning a policy $\hat{\pi}$ via supervised learning on the dataset $\mathcal{D}_L$ (Pomerleau, 1988). Given an hypothesis class $\Pi$ of policies, a standard approach is to minimize the cross-entropy:

$$\hat{\pi}_{\text{BC}} \in \arg\min_{\pi \in \Pi} -\frac{1}{N_L} \sum_{i=1}^{N_L} \log \pi(a_i \mid s_i). \tag{2}$$

The hope is that, given enough samples, we will have that $\hat{\pi}_{\text{BC}} \approx \pi^*$ so that $r(\hat{\pi}_{\text{BC}}) \approx r(\pi^*)$.

### 2.3. IDM-Based Semi-Supervised Imitation Learning

The fact that expert state-action pairs are usually costly to acquire motivated the development of methods that can also leverage action-free transitions of the form $(s, s')$, which are typically much more abundant. In this setting, we are given a small dataset of action-labeled transitions $\mathcal{D}_L := \{(s_i, a_i, s_i')\}_{i=1}^{N_L}$ as well as a very large dataset of unlabeled transitions $\mathcal{D}_U := \{(s_i, s_i')\}_{i=N_L+1}^{N_L+N_U}$, assumed to have been sampled i.i.d. from $p_{\pi^*}(s, a, s')$, defined in (1). We refer to this setting as *semi-supervised imitation learning* (SSIL) (Baker et al., 2022). While our work focuses on learning offline from fixed datasets, we note a number of similar settings where $\mathcal{D}_L$ is built online through environment interactions (Torabi et al., 2018; Radosavovic et al., 2021). The broad field of learning from action-free expert data is often discussed under the umbrella term *Learning from observations (LfO)* (Torabi et al., 2019).

We now review three methods for SSIL that rely on learning an *inverse dynamics model* (IDM) which, given a pair of consecutive states, predicts the action taken. These methods are compared in Table 1.

#### 2.3.1. VM-IDM LEARNING

One approach to leverage this additional unlabeled dataset $\mathcal{D}_U$ is to train a video model (VM) and combine it with an IDM to define a policy.

First, a VM $\hat{v}(s' \mid s)$ is trained to minimize the next state prediction loss $-\frac{1}{N} \sum_{i=1}^N \log \hat{v}(s_i' \mid s_i)$, where $N := N_L + N_U$, thus using all available $(s_i, s_i')$. Second, an IDM $\hat{h}(a \mid s, s')$ is trained to predict the action $a_i$ from the transition $(s_i, s_i')$, using the small action-labeled dataset $\mathcal{D}_L$:

$$\hat{h} \in \arg\min_{h \in \mathcal{H}} -\frac{1}{N_L} \sum_{i=1}^{N_L} \log h(a_i \mid s_i, s_i'), \tag{3}$$

| | | | |
|---|---|---|---|
| **Behavior cloning** | $\min\limits_{\hat{\pi}} -\frac{1}{N_L}\sum_{i=1}^{N_L}\log\hat{\pi}(a_i \mid s_i)$ | | |
| **VM-IDM** | *(1) IDM learning*

$\min\limits_{\hat{h}} -\frac{1}{N_L}\sum_{i=1}^{N_L}\log\hat{h}(a_i \mid s_i, s'_i)$ | *(2) VM learning*
$\min\limits_{\hat{v}} -\frac{1}{N}\sum_{i=1}^{N}\log\hat{v}(s'_i \mid s_i)$ | |
| **IDM labeling** | | *(2) IDM labeling*
$\min\limits_{\hat{\pi}} -\frac{1}{N}\sum_{i=1}^{N}\mathbb{E}_{\hat{h}(a\mid s_i,s'_i)}\log\hat{\pi}(a \mid s_i)$ | |
| **LAPO** | *(1) Reconstruction*


$\min\limits_{\tilde{f},\tilde{h}}\frac{1}{N}\sum_{i=1}^{N}\|s'_i - \tilde{f}(s_i, \tilde{h}(s_i, s'_i))\|^2$ | *(2) LIDM labeling*
$\min\limits_{\tilde{\pi}}\frac{1}{N}\sum_{i=1}^{N}\|\tilde{h}(s_i, s'_i) - \tilde{\pi}(s_i)\|^2$ | *(3) Latent policy decoding*
$\min\limits_{\phi} -\frac{1}{N_L}\sum_{i=1}^{N_L}\log\phi(a_i \mid \tilde{\pi}(s_i))$ |
| **LAPO+** (Ours) | | *(2) LIDM decoding*
$\min\limits_{\phi} -\frac{1}{N_L}\sum_{i=1}^{N_L}\log\phi(a_i \mid \tilde{h}(s_i, s'_i))$ | *(3) IDM labeling*
$\min\limits_{\hat{\pi}} -\frac{1}{N}\sum_{i=1}^{N}\mathbb{E}_{\phi(a\mid\tilde{h}(s_i,s'_i))}\log\hat{\pi}(a \mid s_i)$ |

*Table 1.* Summarizing BC and the four IDM-based semi-supervised imitation learning (SSIL) methods covered in this work, including our contribution, LAPO+ (Section 5.1). SSIL methods leverage action-labeled transitions $(s, a, s')$ from a small dataset $\mathcal{D}_L$ (blue-shaded cells) and a typically much larger dataset $\mathcal{D}_U$ of unlabeled transitions $(s, s')$ (white cells) to learn policies. BC, IDM labeling and LAPO+ directly parametrize a policy $\hat{\pi}$. The VM-IDM policy sequentially samples from $\hat{v}$ then $\hat{h}$ and the LAPO policy is $\phi(a \mid \tilde{\pi}(s))$.

where $\mathcal{H}$ is an hypothesis class of IDMs. These two models define the *VM-IDM policy* from which, given a state $s$, we can sample as follow: (i) sample the next-state $\hat{s}'$ from the VM $\hat{v}(s' \mid s)$, and (ii) sample $\hat{a}$ from the IDM $\hat{h}(a \mid s, \hat{s}')$.

VM-IDM policies are also known as *Video-Action Models (VAM)* (Pai et al., 2025) and have recently been described as *IDM-style Policies* with the VM acting as a world model (Hou et al., 2026). The modular nature of VM-IDM allows to use different datasets or pretrained models for learning/defining $\hat{v}$ and $\hat{h}$. For $\hat{v}$, a core objective is to achieve increased generalization by leveraging internet-scale action-free video data, either to train $\hat{v}$ directly or fine-tune existing video generative models (Du et al., 2023; Liang et al., 2024; Hu et al., 2024; Liang et al., 2024; 2025; Pai et al., 2025). In cases where domain knowledge of the ground-truth IDM is available, methods can leverage specialized pretrained models and algorithms to define $\hat{h}$. Examples include combining pretrained models for vision tasks (e.g., object tracking, optical flows) and inverse kinematics for robot manipulation (Ko et al., 2023; Liang et al., 2024). Additional considerations include interpretability (Liang et al., 2024), flexible goal specification through text-conditioning of the VM, and independence of the VM from a specific action space (Du et al., 2023; Ko et al., 2023). VM-IDM policies can also be trained end-to-end, as shown in the case of *Predictive Inverse Dynamics Models* (PIDM) (Tian et al., 2024).

### 2.3.2. IDM Labeling

Another strategy is to train a policy $\hat{\pi}$ to predict the output of the trained IDM $\hat{h}$ from (3). This is achieved by minimizing the cross-entropy loss $-\frac{1}{N}\sum_{i=1}^{N}\mathbb{E}_{\hat{h}(a\mid s_i,s'_i)}\log\hat{\pi}(a \mid s_i)$ w.r.t. $\hat{\pi}$ alone ($\hat{h}$ is kept frozen), where all available pairs $(s_i, s'_i)$ are used. This can also be understood as labeling the unlabeled transitions $(s_i, s'_i)$ with actions sampled from the

IDM, i.e. $\hat{a}_i \sim \hat{h}(a \mid s_i, s'_i)$, and finally training a policy via BC on the newly labeled transitions.

This simple 2-step formulation of IDM labeling was introduced and shown to work on Minecraft in VPT (Baker et al., 2022) using contractor data for $\mathcal{D}_L$ and unlabeled online gameplay videos for $\mathcal{D}_U$. Variants of IDM labeling have been successfully applied to autonomous driving (Zhang et al., 2022), computer-use agents (Lu et al., 2025), and generated robot videos (Jang et al., 2025).

### 2.3.3. Latent Action Policies

The IDM-based SSIL methods we introduced so far only train the IDM $\hat{h}$ on the action-labeled $\mathcal{D}_L$. Leveraging the larger unlabeled $\mathcal{D}_U$ to learn IDMs has been the subject of recent research efforts leveraging *latent actions* (Edwards et al., 2019; Ye et al., 2022; Bruce et al., 2024; Ye et al., 2024). We specifically focus on the three-stage offline LAPO algorithm (Schmidt & Jiang, 2024).

In a **first stage**, LAPO pretrains jointly a *latent inverse dynamics model* (LIDM) $z = \tilde{h}(s, s')$ and a *latent forward dynamics model* (LFDM) $\hat{s}' = \tilde{f}(s, z)$ to minimize the reconstruction loss $\frac{1}{N}\sum_{i=1}^{N}\|s'_i - \tilde{f}(s_i, \tilde{h}(s_i, s'_i))\|^2$. The variable $z = \tilde{h}(s, s')$ in $\mathcal{Z}$ is meant to be a continuous latent representation of the unobserved action. To induce an information bottleneck, LAPO additionally applies vector quantization to $z$ before passing it to the LFDM (Bengio et al., 2013; Van Den Oord et al., 2017).

In a **second stage**, a *latent action policy* $\tilde{\pi}(s) \in \mathcal{Z}$ is trained to predict the output of the LIDM by minimizing $\frac{1}{N}\sum_{i=1}^{N}\|\tilde{h}(s_i, s'_i) - \tilde{\pi}(s_i)\|^2$ w.r.t. $\tilde{\pi}$ keeping $\tilde{h}$ frozen. Pre-quantized latent actions are used during this stage.

The **third stage** decodes the output of $\tilde{\pi}(s)$ from the latent action space $\mathcal{Z}$ to the true action space $\mathcal{A}$ by minimizing $-\frac{1}{N_L}\sum_{i=1}^{N_L}\log\phi(a_i \mid \tilde{\pi}(s_i))$ w.r.t. to a decoding head

$\phi(a \mid z)$ keeping $\tilde{\pi}$ frozen. The composition $\phi \circ \tilde{\pi}$ forms the final policy $\hat{\pi}$.

We observe that the second stage of LAPO effectively implements "latent IDM labeling" in the latent action space $\mathcal{Z}$, which then provides a fixed pretrained backbone to perform BC during stage three. We will take advantage of this perspective to propose an empirically superior version of LAPO in Section 5.1. Before this, we will spend time connecting the VM-IDM and IDM labeling approaches (Section 3) and discussing particular factors contributing to the sample efficiency of IDM learning in general (Section 4).

## 3. Connecting VM-IDM and IDM labeling

In this section, we show that, in the limit of infinitely many unlabeled transitions, VM-IDM and IDM labeling find the same policy, which we call the *IDM-based policy*. We then show that, when we have infinitely many labeled samples, the IDM-based policy recovers the expert $\pi^*$, just like BC.

We first note that the VM-IDM policy described in Section 2.3.1 can be written as

$$\hat{\pi}_{\hat{v},\hat{h}}(a \mid s) := \int \hat{h}(a \mid s, s')\hat{v}(s' \mid s)ds', \qquad (4)$$

where $\hat{v}$ and $\hat{h}$ are the learned VM and IDM, respectively. To see this, observe that the sampling procedure from Section 2.3.1 corresponds to sampling from $\hat{h}(a \mid s, s')\hat{v}(s' \mid s)$ via *ancestral sampling* (Bishop, 2007, Section 8.1.2). Moreover, the integral marginalizing out $s'$ reflects the fact that $s'$ is discarded once the action is sampled.

**Equivalence between VM-IDM and IDM labeling.** In the VM-IDM method with infinitely many unlabeled pairs $(s, s')$, the VM learning objective stated in Section 2.3.1 becomes $-\mathbb{E}_{p_{\pi^*}(s)}\mathbb{E}_{p_{\pi^*}(s'|s)} \log \hat{v}(s' \mid s)$, which has the same minimizers as

$$\mathbb{E}_{p_{\pi^*}(s)}\mathbb{E}_{p_{\pi^*}(s'|s)} [\log p_{\pi^*}(s' \mid s) - \log \hat{v}(s' \mid s)]$$
$$= \mathbb{E}_{p_{\pi^*}(s)}D_{KL}(p_{\pi^*}(s' \mid s) \parallel \hat{v}(s' \mid s)).$$

It is itself minimized precisely when $\hat{v}(s' \mid s)$ equals the ground-truth VM induced by the expert, defined by $v^*(s' \mid s) := p_{\pi^*}(s' \mid s)$. Note that $v^*(s' \mid s)$ can be derived from $p_{\pi^*}(s, a, s')$ using the definition of a conditional probability distribution/density. The policy obtained by combining this ground-truth VM $v^*$ together with the learned IDM $\hat{h}$ is thus $\hat{\pi}_{v^*,\hat{h}}$, reusing the notation from (4).

In IDM labeling with infinitely many unlabeled transitions, the learning objective from Section 2.3.2 becomes

$$-\mathbb{E}_{p_{\pi^*}(s)v^*(s'|s)\hat{h}(a|s,s')} \log \hat{\pi}(a \mid s),$$

which is equal to $-\mathbb{E}_{p_{\pi^*}(s)\hat{\pi}_{v^*,\hat{h}}(a|s)} \log \hat{\pi}(a \mid s)$. By a similar KL argument, it is minimized precisely when $\hat{\pi} = \hat{\pi}_{v^*,\hat{h}}$.

The last two paragraphs have showed that the policies learned by the VM-IDM approach and by IDM labeling are exactly the same and given by $\hat{\pi}_{v^*,\hat{h}}$ when (i) we have infinitely many unlabeled pairs $(s, s')$, (ii) the hypothesis class for $\hat{v}$ and $\hat{\pi}$ are expressive enough, and (iii) optimization finds a global minimizer. We refer to $\hat{\pi}_{v^*,\hat{h}}$ as the *IDM-based policy*.

**Consistency of the IDM-based policy.** We know that with infinitely many action-labeled samples $(s, a, s')$ and sufficient capacity, the BC policy $\hat{\pi}_{BC}$ becomes equal to the expert policy $\pi^*$ by the same KL argument used in the previous section. It is easy to see that this is also the case for the IDM-based policy. First, let us define the *ground-truth IDM induced by* $\pi^*$ as $h^*(a \mid s, s') := p_{\pi^*}(a \mid s, s')$, which can be derived from $p_{\pi^*}(s, a, s')$, defined in (1). With infinite labeled data and sufficient capacity, we can show via a routine KL argument that IDM learning recovers the ground-truth, i.e. $\hat{h} = h^*$. Consequently, $\hat{h}(a \mid s, s')v^*(s' \mid s) = p_{\pi^*}(a \mid s, s')p_{\pi^*}(s' \mid s) = p_{\pi^*}(a, s' \mid s)$. By integrating w.r.t. $s'$, we get $p_{\pi^*}(a \mid s) = \pi^*(a \mid s)$. In other words, the IDM-based policy learns the expert when given infinite action-labeled data. Note that we just showed $\hat{\pi}_{v^*,h^*} = \pi^*$.

Thus, BC, VM-IDM and IDM labeling are consistent procedures to estimate the expert policy $\pi^*$. However, we will see that IDM-based policy learning can be much more label-efficient than BC.

## 4. Understanding IDM-based policies

Baker et al. (2022) argued that (i) IDM learning (Equation 3) is more sample efficient than BC and that (ii) this contributes to the superiority of IDM labeling policies against BC in SSIL. In this section, we begin by formalizing (i) and (ii), and then investigate specific explanations for (i).

We formalize the claim that "IDM learning is more sample efficient than BC" as follows: for most dataset sizes $N_L$, the following holds with high probability:

$$\mathbb{E}_{p_{\pi^*}(s,s')}D_{KL}(h^* \parallel \hat{h}) < \mathbb{E}_{p_{\pi^*}(s)}D_{KL}(\pi^* \parallel \hat{\pi}_{BC}), \quad (5)$$

where both $\hat{h}$ and $\hat{\pi}_{BC}$ were trained on $N_L$ action-labeled transitions. In other words, the learned IDM $\hat{h}$ is likely closer to $h^*$ than $\hat{\pi}_{BC}$ is to $\pi^*$, when both use the same amount of labeled data. Of course, if the learned IDM $\hat{h}$ is close to $h^*$, we expect the IDM-based policies $\hat{\pi}_{v^*,\hat{h}}$ to be close to $\hat{\pi}_{v^*,h^*} = \pi^*$. We formalize this last point with the following inequality:

$$\mathbb{E}_{p_{\pi^*}(s)}D_{KL}(\pi^* \parallel \hat{\pi}_{v^*,\hat{h}}) \leq \mathbb{E}_{p_{\pi^*}(s,s')}D_{KL}(h^* \parallel \hat{h}), \quad (6)$$

proved in Appendix B. By combining (5) and (6), we obtain, with high probability,

$$\mathbb{E}_{p_{\pi^*}(s)}D_{KL}(\pi^* \parallel \hat{\pi}_{v^*,\hat{h}}) < \mathbb{E}_{p_{\pi^*}(s)}D_{KL}(\pi^* \parallel \hat{\pi}_{BC}),$$

suggesting IDM-based policies will outperform BC in SSIL

In the rest of this section, we provide a more complete explanation of point (i), i.e. why IDM learning is more sample-efficient than BC. While the superior sample efficiency of IDM learning has been observed empirically in Minecraft (Baker et al., 2022) and in the context of VM-IDM policies for robotics (Liang et al., 2025; Pai et al., 2025), it is not entirely clear when and why this should be expected. Moreover, this point might be surprising given the IDM has *twice* as many inputs as the policy, implying a more complex hypothesis class typically leading to worst sample efficiency.

We attribute the observed superior sample efficiency of IDM learning against BC to two causes: First, the ground truth IDM $h^*$ tends to be **less complex** (Section 4.1) and **less stochastic** (Section 4.2) than the expert $\pi^*$. Our analysis provides a more complete explanation for empirical observations and offers a valuable framework for understanding the performance gap between BC and IDM-based policies in different environments, as we will explore in Section 5.

### 4.1. Expert policy is more complex than IDM

In this section, we argue the following claim.

**Claim 1.** *The ground-truth IDM $h^*(a \mid s, s')$ is often **less complex** than the expert $\pi^*(a \mid s)$, and this contributes to IDM learning being more sample-efficient than BC.*

In other words, in many settings, expressing $\pi^*$ requires a more complex/expressive model than what is needed to express $h^*$ which suggests that we can strike a better *bias-variance tradeoff* when learning $h^*$ than when learning $\pi^*$. Indeed, if $h^*$ is simple, it means there is a low complexity hypothesis class $\mathcal{H}$ that contains $h^*$, which results in $\hat{h}$ (Equation 3) being *unbiased*. Moreover, the fact that $\mathcal{H}$ is low complexity means that $\hat{h}$ has *low variance*. In contrast, since $\pi^*$ is complex, it requires a larger hypothesis class $\Pi$ to make sure $\hat{\pi}_{BC}$ (Equation 2) is unbiased, which in turn implies a larger variance and thus poorer generalization.

We investigate Claim 1 on simple maze environments generated and solved using `mazelab` (Zuo, 2018). We study two factors contributing to the complexity of $\pi^*$, namely *environment complexity* and *goal complexity*.

#### 4.1.1. ENVIRONMENT COMPLEXITY

**Experimental setup (Figure 1).** We generate the transitions $(s, a, s') \in \mathcal{D}_L$ using the maze environment and an expert policy $\pi^*$ solving the maze deterministically. We compare the test accuracy of BC and VM-IDM along four axes: *maze complexity*, *train split*, *state space format* & *model capacity*. *Maze complexity* refers to the size of the maze (10x10, 20x20 or 50x50). *Train split* refers to the proportion of states $s$ that

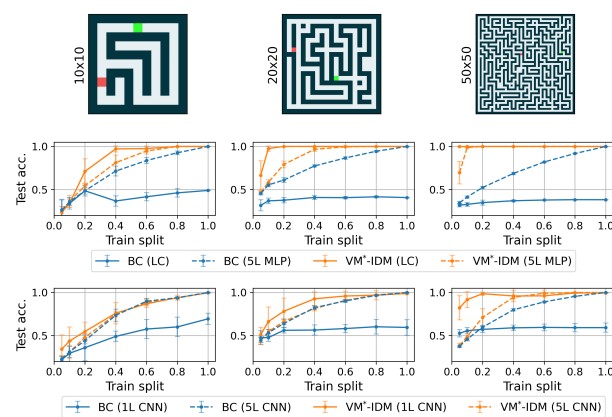

*Figure 1.* **Varying the environment complexity.** Comparing BC and VM*-IDM, with different architectures for $\hat{\pi}$ and $\hat{h}$, respectively: *LC* is a linear classifier, *5L MLP* is a 5-layer multilayer perceptron and *nL CNN* is an $n$-layer convolutional neural network. See Appendix D.1 for details. Averaging over 5 seeds.

are present in the action-labeled dataset $\mathcal{D}_L$, where 1 means the full state space is sampled. The *state space format* is either $\mathcal{S}^{pos}$, where the state is the $(x, y)$ position of the agent, or $\mathcal{S}^{img}$, where the state is an image representation of the maze including the player and the goal. *Model capacity* takes two values, either *high* or *low*: When the state space format is $\mathcal{S}^{pos}$, $\mathcal{H}_{low}$ and $\Pi_{low}$ are linear classifiers (LC), while $\mathcal{H}_{high}$ and $\Pi_{high}$ are multilayer perceptrons (MLP). When the state space format is $\mathcal{S}^{img}$, $\mathcal{H}_{low}$ and $\Pi_{low}$ are one-layer convolutional networks (1L CNN) and $\mathcal{H}_{high}$ and $\Pi_{high}$ are 5-layers CNNs (5L CNN). In all cases, the goal location is fixed across episodes (see Section 4.1.2 for experiments involving multiple goals). Note that, throughout Section 4.1, the VM-IDM uses the ground-truth video model $v^*$, echoing the infinite $\mathcal{D}_U$ regime. For more details, see Appendix D.1.

**Results (Figure 1).** First, we see that for each maze environment and each state space format, the low capacity VM-IDM achieves perfect test accuracy when given enough samples (indicating $h^* \in \mathcal{H}_{low}$), while low-capacity BC policy never reaches perfect accuracy (indicating $\pi^* \notin \Pi_{low}$). This suggests that $h^*$ is less complex than $\pi^*$. Secondly, while both low-capacity VM-IDM and high-capacity BC reach perfect performance when seeing enough data, the former outperforms the latter in the low data-regime; and this gap increases for more complex mazes. This suggests IDM learning is more sample-efficient than BC in this setting. Indeed, since $h^*$ is simpler than $\pi^*$, we can use a smaller architecture to reduce variance without introducing bias, resulting in better test accuracy. Lastly, we observe that the high-capacity VM-IDM outperforms the high-capacity BC policy, especially for the more complex mazes. This trend is also present in the image state format, although not as strongly as for the position state format. We attribute this to the simplicity bias of neural networks which, while very expressive, have an implicit bias towards simpler functions.

We will come back to this point soon.

**Simple analytic form for $h^*$ but not for $\pi^*$.** We now show that the ground-truth IDM $h^*$ can be expressed as a linear classifier, while the expert $\pi^*$ cannot. In a maze, executing an action deterministically moves the agent in the corresponding direction by one unit unless a wall prevents this move. Since the expert policy $\pi^*$ never runs into a wall, its induced ground-truth IDM $h^*$ takes a particularly simple form. Indeed, this can be seen by noticing that, when avoiding walls, we can exactly infer the action taken from the state difference $s' - s$. For example, when $s' - s = (1,0)^\intercal$, we can infer $a = \texttt{right}$; when $s' - s = (-1,0)^\intercal$, we can infer $a = \texttt{left}$; and so on. Since the points $\{(1,0),(-1,0),(0,1),(0,-1)\}$ are linearly separable and because the map $(s,s') \mapsto s' - s$ is linear, we can express $h^*(a \mid s,s')$ as $\mathrm{softmax}_a(V(s,s'))$ for some matrix $V \in \mathbb{R}^{4 \times 4}$. We make this construction explicit in Appendix C.1. In other words, $h^* \in \mathcal{H}_{\mathrm{low}}$, i.e. $\hat{h}$ is unbiased. This explains why the low-capacity VM-IDM obtains perfect test accuracy in Figure 1 when trained on sufficiently many labeled samples. Now contrast this with the expert policy $\pi^*(a \mid s)$. For complex mazes, there is no obvious simple rule mapping the state $s$ to the optimal action $a$. At the very least, it cannot be expressed by a linear classifier, as was the case for $h^*$. The expert $\pi^*$ has to somehow "memorize" all the trajectories leading to the end of the maze. In this case, the expert policy $\pi^*$ is much more complex than its corresponding ground-truth IDM $h^*$. The story is similar when the state $s$ is an image of the maze: We can formally show that the IDM $h^*$ can be expressed with a simple single layer convolutional neural network (see Appendix C.2) while it appears the policy cannot (as suggested by its inferior performance in Figure 1). In most realistic settings, we do not expect to always be able to write down a simple analytic form for the ground-truth IDM (otherwise, why would one learn it from data?) but we believe these examples serve as useful illustrations capturing the essence of our argument.

**Generalization via capacity control.** *Capacity control* as a method to reduce variance is well-known: Reducing the size of the hypothesis class reduces the variance of the predictor. This common wisdom is formalized by the classical literature on statistical learning theory (Shalev-Shwartz & Ben-David, 2014; Mohri et al., 2018) which proposes quantitative measures of complexity for hypothesis classes which upperbound, with high probability, the difference between the generalization loss and the empirical loss of a learned predictor. Measures of complexity include the VC dimension (Vapnik & Chervonenkis, 1971) as well as the Rademacher complexity (Koltchinskii & Panchenko, 2000; Bartlett & Mendelson, 2003). These generalization bounds establish that controlling the capacity/expressivity of an hypothesis class reduces the variance of the learned predictor.

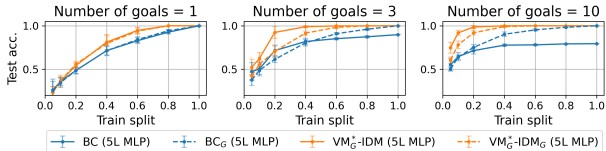

*Figure 2.* **Varying the number of goals.** $\mathrm{BC}_G$ is behavior cloning with goal conditioning, $\mathrm{VM}^*_G$-IDM is $\mathrm{VM}^*$-IDM where the VM is goal conditioned and the IDM is not; and $\mathrm{VM}^*_G$-$\mathrm{IDM}_G$ is when both the VM and the IDM are goal conditioned. See Appendix D.1 for details. Averaging over 5 seeds.

This explains why the low-capacity VM-IDM policy (which we just showed is unbiased) outperforms the high-capacity one in the low-data regime.

**Generalization via simplicity bias of NN.** Note that capacity control cannot explain why, in Figure 1, the high-capacity VM-IDM policy outperforms the high-capacity BC policy in the low-data regime, since both $\hat{h}$ and $\hat{\pi}_{\mathrm{BC}}$ are implemented with the same 5-layer neural network. Even worse, we have in fact that the complexity of $\mathcal{H}_{\mathrm{high}}$ is greater than that of $\Pi_{\mathrm{high}}$, since the IDM has twice as many inputs as the policy, which suggests that $\hat{\pi}_{\mathrm{BC}}$ should generalize better than $\hat{\pi}_{v^*,\hat{h}}$, contrary to our observation. We explain this observation by the *simplicity bias of neural networks*: When a neural network is in the *overparametrized interpolation regime*, i.e. when multiple parameter configurations achieve zero training loss, training tends to favor the interpolating functions which are simpler/smoother. We hypothesize that this simplicity bias favors the simpler $h^*$ over the more complex $\pi^*$, thus explaining why fewer samples are needed to achieve greater performance for the VM-IDM policy. While the reasons why neural networks are implicitly biased towards simpler functions are not fully understood, theoretical explanations have been proposed, many of which are based on properties of gradient-based optimization such as potential regularizing effects (Advani & Saxe, 2017; Soudry et al., 2018; Gidel et al., 2019) and its tendency to first learn linear classifiers before learning increasingly complex functions (Kalimeris et al., 2019). Other works attribute the simplicity bias to the parameter-to-function map of neural architectures (Valle-Pérez et al., 2019; Mingard et al., 2021; Chiang et al., 2023; Buzaglo et al., 2024; Dziugaite & Roy, 2025).

**Stochastic environment.** In Appendix D.1.4, we explore a stochastic variant of this setting where the agent remains static with some probability, regardless of the action taken. Interestingly, the ground-truth IDM $h^*$ is not linear in this setting. We nevertheless reach similar conclusions.

### 4.1.2. GOAL COMPLEXITY

In the last section, we argued that the complexity of the environment impacts the complexity of the expert $\pi^*$ more

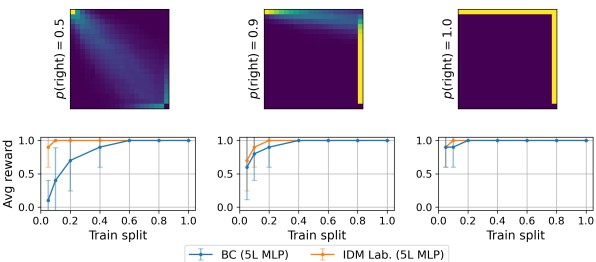

*Figure 3.* **Varying the stochasticity of the expert.** First row shows the state visitation distributions for different experts. Comparing the average reward of BC and IDM labeling. See Appendix D.1 for details. Averaging over 10 seeds.

than it impacts the complexity of the IDM $h^*$, leading to IDM learning having a statistical advantage over BC. We now argue that the *diversity of goals* also behaves similarly.

**Experimental setup (Figure 2).** We consider the same 10x10 maze environment as in Section 4.1.1, but this time the goal location changes from one trajectory to another. We assume the goal $(x, y)$-location, $g$, is observed with each transition $(s, a, s')$ allowing *goal-conditioning* for each model. We explore the impact of goal-conditioning for the BC and VM-IDM policies. We look at the BC policy with and without goal-conditioning, that is with and without $g$ as input. We also look at the VM-IDM policy $\hat{\pi}_{v^*, \hat{h}}$ where the VM is always goal-conditioned and equal to the ground-truth, i.e. $\hat{v}(s' \mid s, g) := p_{\pi^*}(s' \mid s, g)$, and where the IDM is either $\hat{h}(a \mid s, s')$ or $\hat{h}(a \mid s, s', g)$.

**Results (Figure 2).** First, we see without surprise that BC without goal-conditioning cannot reach perfect test accuracy even when seeing all possible transitions $(s, a, s')$, i.e. it is biased. In contrast, VM-IDM without goal-conditioning does reach perfect test accuracy, indicating that the goal $g$ is not necessary to predict the action given $(s, s')$, which of course makes intuitive sense in this simple environment. This is thus another way in which the IDM $h^*$ can be less complex than the expert $\pi^*$, because the former effectively does not depend on $g$ while the latter does. While BC with goal-conditioning does reach perfect accuracy with enough samples, in the low-data regime it is outperformed by VM-IDM, especially without goal-conditioning. This shows how IDM learning is more sample-efficient than BC in this setting.

### 4.2. Expert policy is more stochastic than IDM

In this section, we argue the following claim.

**Claim 2.** *The ground-truth IDM $h^*(a \mid s, s')$ is often **less** stochastic than the expert $\pi^*(a \mid s)$, and this contributes to IDM learning being more sample efficient than BC.*

In general, we always have $H(a \mid s, s') \leq H(a \mid s)$,

where $H(a \mid s) := -\mathbb{E}_{s,a} \log \pi^*(a \mid s)$ and $H(a \mid s, s') := -\mathbb{E}_{s,a,s'} \log h^*(a \mid s, s')$ are *conditional entropies*. We claim that this inequality is often strict, meaning $h^*(a \mid s, s')$ is less stochastic on average than $\pi^*(a \mid s)$. This can happen, e.g., when multiple actions are equally optimal, allowing for a stochastic expert; and when observing $(s, s')$ reveals with high certainty the action taken, which is the case in many navigation tasks (e.g. maze environments). This higher stochasticity makes BC less sample-efficient than IDM learning because the sample-efficiency of an estimator is often negatively impacted by the stochasticity of the ground-truth data-generating process (DGP). For example, in linear regression, the variance of the least-square estimator $\hat{\beta}$ is proportional to the variance of the residual noise $\epsilon$ in the DGP $y = \beta^* x + \epsilon$, under the Gauss-Markov conditions (Sen & Srivastava, 1991, Section 1.8). Also, generalization bounds for binary classification show how having a "likely not too stochastic" DGP $p^*(y \mid x)$ leads to improved sample efficiency (Mammen & Tsybakov, 1999).[1] We now investigate Claim 2 on a maze-like environment.

**Experimental setup (Figure 3).** We consider a simple 20x20 grid environment where the agent starts at the top left corner and must reach the bottom right corner. There are no obstacles preventing movement, except for the borders of the grid. We consider three expert policies to generate data. All three are optimal and consist in going right with probability $p(\texttt{right})$ and going down with probability $1 - p(\texttt{right})$, except on the right or bottom border, where the policy deterministically chooses the only optimal action. We vary $p(\texttt{right})$ to study the effect of the expert stochasticity on the performance of BC and IDM labeling. In both cases, $\hat{\pi}$ and $\hat{h}$ use the same MLP architecture. Note that the induced IDM $h^*$ is always deterministic, contrary to $\pi^*$.

**Results (Figure 3).** In the low-data regime, we see that, as the stochasticity of the expert increases, the reward obtained by BC diminishes, contrarily to the IDM-based policy which is much less impacted. This is because the ground-truth IDM $h^*$ remains deterministic, no matter how stochastic $\pi^*$ is. This suggests the lower stochasticity of $h^*$ contributes to its sample-efficiency.

## 5. Methods and Experiments

The experiments of this section serve a dual purpose.

First, we aim to compare IDM-based policies, BC, and recent baselines for SSIL over a range of architectures and tasks, including Atari-like games with discrete actions (Section 5.1) and manipulation tasks with continuous actions (Section 5.2). When IDM-based policies outperform BC, we discuss if those gains can be be plausibly attributed to

---

[1]We are referring to the Tsybakov noise condition (Mammen & Tsybakov, 1999). See Rigollet (2015), Section 3.2).

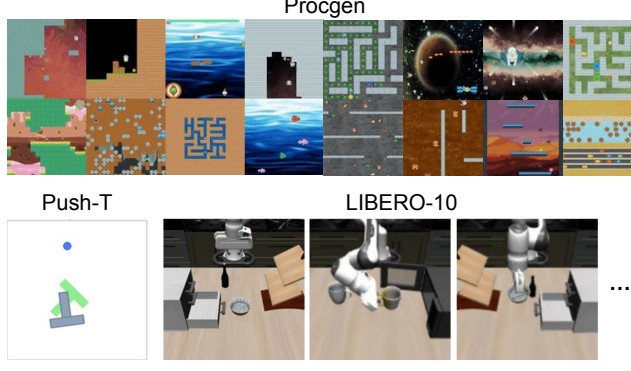

Procgen

Push-T          LIBERO-10

*Figure 4.* Environments for the vision-based experiments in Section 5. **(Top)** Procgen (Cobbe et al., 2019) consists of 16 Atari-like environments with discrete actions. We use the data and settings from LAPO (Schmidt & Jiang, 2024). **(Bottom Left)** Push-T (Chi et al., 2025) requires pushing a T-shaped block (gray) in a target position (green) with a circular end-effector (blue). **(Bottom Right)** LIBERO-10 (Liu et al., 2023) is a set of 10 long-horizon robot manipulation tasks. We use continuous action chunks for both Push-T and LIBERO-10.

the factors discussed in Section 4.

Second, we explore how the observed statistical advantage of IDM learning can be leveraged to improve other IDM-based SSIL methods. We will therefore begin each experiment section by introducing these improvements. In particular, we propose LAPO+, an improved algorithm for latent action policy learning (Section 5.1), and discuss a new VM-IDM sampling scheme for unified video-action models (Section 5.2). Code for all our experiments is available here.

## 5.1. Procgen

**LAPO+.** In Section 2.3.3, we saw that LAPO operates in three stages which are to (1) train a latent inverse dynamics model (LIDM) $\tilde{h}$ and a latent forward dynamics model (LFDM) $\tilde{f}$ by minimizing a reconstruction loss, (2) train a latent policy $\tilde{\pi}$ on transitions labeled by the LIDM, and (3) train a decoding head $\phi$ on top of the latent policy $\tilde{\pi}$ using $\mathcal{D}_L$ to go from the latent actions in $\mathcal{Z}$ to true actions in $\mathcal{A}$. See Table 1 for a detailed summary.

Note that the third stage can be thought of as performing *BC with a pretrained backbone* $\tilde{\pi}$. Motivated by the observation that IDM learning is often more sample-efficient than BC, we propose an alternative to LAPO, coined **LAPO+**, which learns a decoding head $\phi$ on top of the LIDM $\tilde{h}$ to go from latent actions in $\mathcal{Z}$ to true actions in $\mathcal{A}$, corresponding to *IDM learning with a pretrained backbone* $\tilde{h}$.

LAPO and LAPO+ are compared in Table 1. Specifically, LAPO+ uses the same **first stage** as LAPO to learn the LIDM $z = \tilde{h}(s, s')$ and considers a **second stage** where it learns a decoding head $\phi(a \mid z)$ on top of the latent IDM by minimizing $-\frac{1}{N_L} \sum_{i=1}^{N_L} \log \phi(a_i \mid \tilde{h}(s_i, s_i'))$ w.r.t. to $\phi$ and

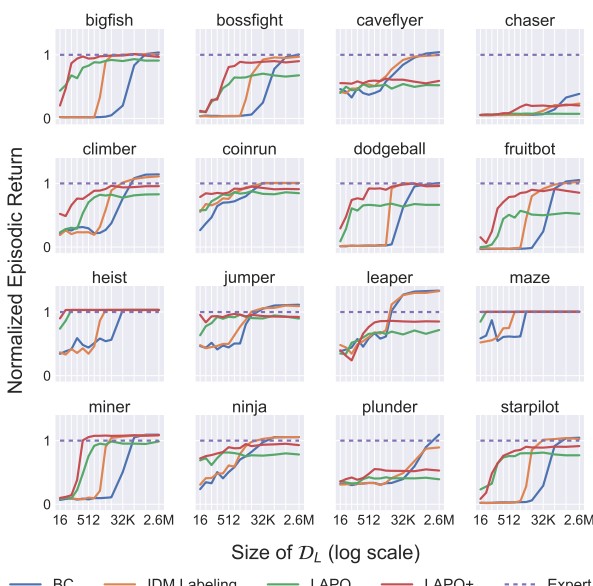

*Figure 5.* Normalized maximum return achieved during training by BC, IDM labeling, LAPO and LAPO+ on the Procgen benchmark (Figure 4), varying the number of action-labeled transitions in $\mathcal{D}_L$. We show the average of 3 seeds, with $\mathcal{D}_L$ being randomly sampled from the whole datasets each time. IDM labeling, LAPO and LAPO+ use all 2.6M transitions without action labels as $\mathcal{D}_U$.

keeping $\tilde{h}$ frozen, resulting in a decoded IDM $\hat{h} = \phi \circ \tilde{h}$. In the **third stage**, it learns the policy $\hat{\pi}$ by performing IDM labeling (Section 2.3.2) using labels generated from $\hat{h} = \phi \circ \tilde{h}$. Conceptually, LAPO+ inverts the second stage (LIDM labeling) and the third stage (decoding $\mathcal{Z}$ to $\mathcal{A}$) of LAPO.

**Results (Figures 5 & 6).** We report the performance of BC, IDM labeling, LAPO and LAPO+ in Procgen environments in Figure 5. Training details are in Appendix D.2.1. We find IDM labeling to outperform BC by a few orders of magnitude in 9 environments, while gains are modest or non-existent on the rest. In Figure 6, we discuss how the factors presented in Section 4 can shed light on the fluctuating sample efficiency gap between IDM labeling and BC.

For their part, LAPO & LAPO+ frequently yield capable policies vastly outperforming BC and IDM labeling in low-data regimes, with LAPO+ consistently outperforming LAPO. This underscores the importance of choosing the right latent function to decode with a small $\mathcal{D}_L$. Somewhat orthogonal to our claims, we note that LAPO and LAPO+ tend to underperform in high-data regimes, suggesting potential trade-offs associated with latent action policies.

## 5.2. Manipulation

**Video-Action Multitasking.** Recent methods for BC use unified, typically high-capacity architectures for video and

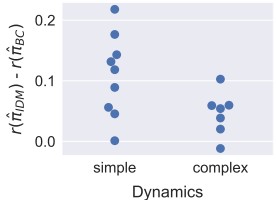 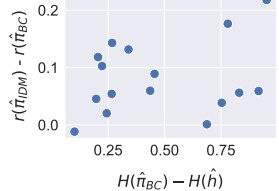

*(a)* Complexity Analysis      *(b)* Stochasticity Analysis

*Figure 6.* We study the complexity and stochasticity factors from Section 4 across the 16 Procgen environments. As an aggregate score, we consider the *mean return gap* $r(\hat{\pi}_{IDM}) - r(\hat{\pi}_{BC})$ between IDM labeling and BC, which we average over $\mathcal{D}_L$ sizes (Figure 5). **Complexity Analysis.** We divide environments based on their dynamics (Appendix D.2.2): *simple* environments are those in which the agent moves strictly because of an input action while the agent can move due to other factors in *complex* environments (e.g., momentum, gravity, moving platform). We expect the IDM to be simpler under simple dynamics and find the mean return gap to be higher in those environments. **Stochasticity Analysis.** As a proxy for the entropy of the ground truth policy and IDM in each environment, we consider the average conditional entropies on a test set of $\hat{\pi}_{BC}$ and the IDM $\hat{h}$ learned on all 2.6M transitions, allowing us to study an approximate entropy gap $H(\hat{\pi}_{BC}) - H(\hat{h})$. We first note that $H(\hat{\pi}_{BC}) > H(\hat{h})$ across all environments, as hinted in Section 4.2. We then observe a positive, although imperfect correlation with the mean return gap.

action prediction where the VM and policy share most of their parameters (Wu et al., 2023; Guo et al., 2024). Follow-up works extend parameter sharing to IDMs and forward dynamics models (FDM), and rely on techniques such as input masking (Li et al., 2025) or separate diffusion timesteps (Zhu et al., 2025) to specifically sample the policy, the VM, the IDM or the FDM. We will refer to this setting as **multi-tasking**. Our experiments focus on UVA (Li et al., 2025), which has shown promising results on robot manipulation. We use the UVA architecture as a testbed to study BC and IDM-based policies using modern features such as diffusion action heads, frame stacking and action chunking. We emphasize that UVA uses a single architecture to implement the policy, VM, IDM and FDM.

**Methods.** We learn **BC**, **IDM labeling** and **VM-IDM** policies using the UVA architecture (without multitasking). We also train standard UVA (with multitasking) which samples actions from its policy (**Policy (UVA)**). As an alternative, we propose to sample actions as a VM-IDM (**VM-IDM (UVA)**). The latter is a compact parametrization of a VM-IDM and we hypothesize that it may lead to some improvements in the low-data regimes (Section 4). We follow the UVA recommendation to pretrain all models on the VM task on $\mathcal{D}_U$, meaning all models in this section perform SSIL via their initialization. Training details are provided in Appendix D.3.

**Results (Figure 7).** We compare these methods on Push-T

and LIBERO-10. IDM-based policies outperform BC in most settings and by an especially large margin on Push-T, something we attribute to the particularly simple nature of the IDM on this problem (the action is the location of the agent). We note that IDM labeling tends to trail VM-IDM by a small margin. We speculate that this could be explained by the relatively limited size of $\mathcal{D}_U$ (as opposed to the infinite regime considered in Section 3) and some generalization abilities of the VM. We finally observe that sampling the same UVA models as VM-IDMs (VM-IDM (UVA)) leads to equal or improved performance over sampling them as policies (Policy (UVA)), illustrating how UVA learns capable IDMs from relatively few samples.

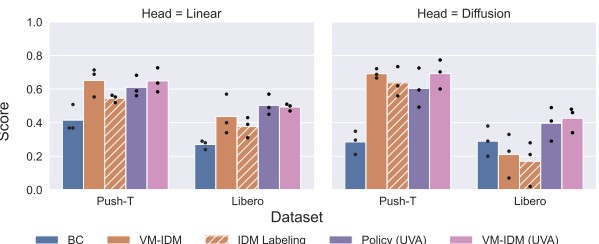

*Figure 7.* Policy performance on Push-T and LIBERO-10 (Figure 4). For $\mathcal{D}_L$, we use 10 demonstrations for Push-T and 2 demonstrations per task for LIBERO-10, representing approximately 5% of available demonstrations, which are all included without action labels in $\mathcal{D}_U$. The Push-T score is the average max IoU with the target T over 50 rollouts while the LIBERO-10 score is the success rate over 100 rollouts. We report the performance of the last checkpoints averaged over 3 training seeds (we resample $\mathcal{D}_L$ for each seed) and overlay the actual score of each seed as black markers. We also report results of a UVA variant with a linear action head (Appendix D.3).

## 6. Discussion

While the unifying concept of IDM-based policy (Section 3) and its formalism (Section 4) provide a valuable framework to understand the empirical performance of IDM-based policies in SSIL settings, some limitations remain. First, our modeling and experiments assume a curated $\mathcal{D}_U$ which only includes expert data sampled precisely from the task of interest. A more practical setting would be when $\mathcal{D}_U$ is sampled from an environment/expert that differs from the one of interest used to sample $\mathcal{D}_L$. Second, we focus on generalization and ignore sampling costs, with VM-IDM approaches being generally slower (although recent research efforts aim to address this shortcoming (Hu et al., 2024; Pai et al., 2025)). Finally, our work focuses on the sample efficiency of IDM learning w.r.t. action labels. We did not investigate the difference in generalization between VM-IDM and IDM labeling in the finite $\mathcal{D}_U$ setting, which might be crucial to fully understand the differences between both methods. These present promising avenues for future work.

## Acknowledgements

The work at the Université de Montréal was supported by a Natural Sciences and Engineering Research Council of Canada (NSERC) PGS D Scholarship (Morin). The authors would like to thank Simon Lacoste-Julien for engaging in insightful discussions that ultimately improved this work. This research was enabled in part by computational resources provided by Calcul Québec (calculquebec.ca), the Digital Research Alliance of Canada (alliancecan.ca) and Mila (mila.quebec).

## Impact Statement

This paper presents work whose goal is to advance the field of Machine Learning. There are many potential societal consequences of our work, none which we feel must be specifically highlighted here.

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

# A. Contribution Statement

**Sacha Morin** and **Sébastien Lachapelle** jointly conceptualized the project, engaging in extensive discussions on the mechanisms behind the success of IDM learning and IDM-based policies. They also authored the manuscript.

**Sacha** led the project and leveraged his knowledge of the SSIL and robotics literature to identify IDM learning as a recurring pattern in a number of recent methods, providing the foundation for the paper. Sacha also designed and ran the Procgen, Push-T, and LIBERO-10 experiments in Section 5.

**Moonsub Byeon** contributed to multiple brainstorming sessions, often providing valuable advice on experiment design.

**Alexia Jolicoeur-Martineau** offered guidance on training video models and provided valuable feedback early on during the project.

**Sébastien** was the lead advisor on this project, contributing key statistical learning perspectives and providing the formalism needed to unify IDM-based methods. Sébastien also led the maze experiments in Section 4.

# B. Proof of Inequality 6

We now show that

$$\mathbb{E}_{p_{\pi^*}(s)} D_{KL}(\pi^*(a \mid s) \parallel \hat{\pi}_{v^*, \hat{h}}(a \mid s)) \leq \mathbb{E}_{p_{\pi^*}(s, s')} D_{KL}(h^*(a \mid s, s') \parallel \hat{h}(a \mid s, s'))$$

**Proof**

$$
\begin{aligned}
\mathbb{E}_{p_{\pi^*}(s,s')} D_{KL}(h^*(a \mid s, s') \parallel \hat{h}(a \mid s, s')) &= \mathbb{E}_{p_{\pi^*}(s,s')} \sum_{a \in \mathcal{A}} h^*(a \mid s, s') \log \frac{h^*(a \mid s, s')}{\hat{h}(a \mid s, s')} \\
&= \mathbb{E}_{p_{\pi^*}(s)} \int \sum_{a \in \mathcal{A}} v^*(s' \mid s) h^*(a \mid s, s') \log \frac{h^*(a \mid s, s')}{\hat{h}(a \mid s, s')} ds' \\
&= \mathbb{E}_{p_{\pi^*}(s)} \int \sum_{a \in \mathcal{A}} v^*(s' \mid s) h^*(a \mid s, s') \log \frac{h^*(a \mid s, s') v^*(s' \mid s)}{\hat{h}(a \mid s, s') v^*(s' \mid s)} ds' \\
&= \mathbb{E}_{p_{\pi^*}(s)} \int \sum_{a \in \mathcal{A}} p_{\pi^*}(a, s' \mid s) \log \frac{p_{\pi^*}(a, s' \mid s)}{\hat{h}(a \mid s, s') v^*(s' \mid s)} ds' .
\end{aligned}
$$

Define $p_{v^*, \hat{h}}(a, s' \mid s) := \hat{h}(a \mid s, s') v^*(s' \mid s)$. From this we obtain

$$
\begin{aligned}
& \mathbb{E}_{p_{\pi^*}(s,s')} D_{KL}(h^*(a \mid s, s') \parallel \hat{h}(a \mid s, s')) \\
&= \mathbb{E}_{p_{\pi^*}(s)} \int \sum_{a \in \mathcal{A}} p_{\pi^*}(a, s' \mid s) \log \frac{p_{\pi^*}(a, s' \mid s)}{p_{v^*, \hat{h}}(a, s' \mid s)} ds' \\
&= \mathbb{E}_{p_{\pi^*}(s)} D_{KL}(p_{\pi^*}(a, s' \mid s) \parallel p_{v^*, \hat{h}}(a, s' \mid s)) \\
&= \mathbb{E}_{p_{\pi^*}(s)} D_{KL}(p_{\pi^*}(a \mid s) \parallel p_{v^*, \hat{h}}(a \mid s)) + \mathbb{E}_{p_{\pi^*}(s,a)} D_{KL}(p_{\pi^*}(s' \mid s, a) \parallel p_{v^*, \hat{h}}(s' \mid s, a)) \\
&= \mathbb{E}_{p_{\pi^*}(s)} D_{KL}(\pi^*(a \mid s) \parallel \hat{\pi}_{v^*, \hat{h}}(a \mid s)) + \mathbb{E}_{p_{\pi^*}(s,a)} D_{KL}(p(s' \mid s, a) \parallel p_{v^*, \hat{h}}(s' \mid s, a)) \\
&\geq \mathbb{E}_{p_{\pi^*}(s)} D_{KL}(\pi^*(a \mid s) \parallel \hat{\pi}_{v^*, \hat{h}}(a \mid s)) ,
\end{aligned}
$$

where we leveraged the chain rule for the KL divergence in the third equality and the fact that the KL divergence is always greater or equal to zero in the last inequality. ∎

# C. Explicit expressions for simple ground-truth IDM in Figure 1

## C.1. Positions as states ($\mathcal{S}_{\text{pos}}$)

We now show that the ground-truth IDM $h^*$ from Figure 1 with position states ($\mathcal{S}_{\text{pos}}$) can be expressed as a linear classifier. Executing an action deterministically moves the player in the corresponding direction by one unit unless a wall prevents

this move. Assuming the expert policy $\pi^*$ never runs into a wall (which is the case here), its induced ground-truth IDM $h^*$ takes a particularly simple form. Indeed, this can be seen by noticing that, when avoiding walls, we can exactly infer the action taken from the state difference $s' - s$. For example, when $s' - s = (1, 0)^\intercal$, we can infer $a = \texttt{right}$; when $s' - s = (-1, 0)^\intercal$, we can infer $a = \texttt{left}$; and so on. Moreover, the map from $s' - s$ to $a$ can be expressed with a simple linear classifier $\mathrm{softmax}(\tau^{-1} W(s' - s))$ where $\tau$ is a temperature parameter and

$$W := \begin{bmatrix} 1 & -1 & 0 & 0 \\ 0 & 0 & 1 & -1 \end{bmatrix}^\intercal.$$

Importantly, when $\tau \to 0$, we have that $\mathrm{softmax}_a(\tau^{-1} W(s' - s))$ approaches the ground-truth IDM $h^*(a \mid s, s')$, since it concentrates probability mass on the correct action. Since the map $(s, s') \mapsto \tau^{-1} W(s' - s)$ is linear, we have that $h^*$ can be (approximately) expressed by a linear classifier. In other words, $h^*$ is in the closure of $\mathcal{H}_{\mathrm{low}}$.

### C.2. Images as states ($\mathcal{S}_{\mathrm{img}}$)

We now show that the ground-truth IDM $h^*$ from Figure 1 with image states ($\mathcal{S}_{\mathrm{img}}$) can be expressed as a simple one-layer network with a convolutional layer with a 3x3 kernel, mapping the 6 input channels ($s$ and $s'$ have three channels each) to 4 channels (one per action) without padding nor stride; with a global max pooling layer (same as described in D.1.1). For each action $a$, we construct the 3x3 kernel $K^a \in \mathbb{R}^{6 \times 3 \times 3}$. Let $x, x' \in \mathbb{R}^{3 \times 3 \times 3}$ be 3x3 patches respectively from $s$ and $s'$ taken at the same location. We can see $K^a$ as the concatenation of two kernels, namely $W^a, V^a \in \mathbb{R}^{3 \times 3 \times 3}$. The dot product between the kernel $K^a = (W^a, V^a)$ and the patch $(x, x')$ can thus be written as

$$\langle K^a, (x, x') \rangle = \langle W^a, x \rangle + \langle V^a, x' \rangle$$

By setting $W^a := -V^a$ we get

$$\langle K^a, (x, x') \rangle = \langle -V^a, x \rangle + \langle V^a, x' \rangle = \langle V^a, -x \rangle + \langle V^a, x' \rangle = \langle V^a, x' - x \rangle.$$

Let $(x_{\mathrm{right}}, x'_{\mathrm{right}})$ be a pair of patches taken at the same location in subsequent images $s, s'$ where the agent is located at the center of the patch $x_{\mathrm{right}}$ and move one step to the right in patch $x'_{\mathrm{right}}$. The specific choice of transition $(s, s')$ and patch location will not matter, as long as the agent is centered in the patch $x_{\mathrm{right}}$ and moved to the right in $x'_{\mathrm{right}}$. We can select pairs $(x_{\mathrm{left}}, x'_{\mathrm{left}})$, $(x_{\mathrm{up}}, x'_{\mathrm{up}})$ and $(x_{\mathrm{down}}, x'_{\mathrm{down}})$ analogously. Then, we can define $V^a$ as

$$V^{\mathrm{right}} := x'_{\mathrm{right}} - x_{\mathrm{right}}, \quad V^{\mathrm{left}} := x'_{\mathrm{left}} - x_{\mathrm{left}}, \quad V^{\mathrm{up}} := x'_{\mathrm{up}} - x_{\mathrm{up}}, \quad V^{\mathrm{down}} := x'_{\mathrm{down}} - x_{\mathrm{down}}.$$

Note that the precise choice of patch localization for $x_a$ and $x'_a$ does not matter because in the end the resulting difference $x'_a - x_a$ will remain the same.

Now observe that at most two pixels change between any pair of patches $(x, x')$ taken from any feasible transition $(s, s')$. Also, every pixel change swaps the agent pixel and a background pixel or vice-versa (we ignore the goal pixel for simplicity), contributing an equal amount of $p$ to the magnitude $\|x' - x\|^2$, where $p$ is the squared norm of the difference between the "agent pixel" and the "background pixel" (a pixel is a 3-dimensional vector for RGB). We therefore always have

$$\|x' - x\| \le \sqrt{2p}.$$

Fix some action $a$. For all patch transition $(x, x')$, we have that $\langle V^a, x' - x \rangle \le \|V^a\| \|x' - x\|$ by Cauchy-Schwarz inequality, which implies for all $(x, x')$

$$\langle V^a, x' - x \rangle \le \|V^a\| \|x' - x\| \le \sqrt{2p} \sqrt{2p} = 2p.$$

Note that $\|x'_a - x_a\| = \sqrt{2p}$ since one pixel changed from background to agent and one pixel changed from agent to background. This means

$$\langle V^a, x'_a - x_a \rangle = \langle x'_a - x_a, x'_a - x_a \rangle = \|x'_a - x_a\|^2 = 2p,$$

and thus $z_a := x'_a - x_a$ is the unique maximizer of $\langle V^a, z \rangle$ subject to $\|z\| \le \sqrt{2p}$. Uniqueness can be seen as follows: Another maximizer $\tilde{z}$ would have to achieve $2p = \langle V^a, \tilde{z} \rangle \le \|V^a\| \|\tilde{z}\| \le 2p$ which means the Cauchy-Schwarz inequality is actually an equality and thus $V^a$ and $\tilde{z}$ would have to be linearly dependent (Cauchy-Schwarz) with equal norm (since $\|V^a\| \|\tilde{z}\| = 2p$) and equal direction, i.e. $\tilde{z} = x'_a - x_a$.

Consider a feasible transition $(s, a^*, s')$ and all the patch differences $x'_{i,j} - x_{i,j}$ taken from $(s, s')$ at location $(i, j)$. Since $x'_{a^*} - x_{a^*}$ will appear exactly at one location $(i^*, j^*)$ (the expert $\pi^*$ always moves and the agent is unique) and $\|x'_{i,j} - x_{i,j}\| \le \sqrt{2p}$, we must have that

$$\langle V^{a^*}, x'_{i,j} - x_{i,j} \rangle < \langle V^{a^*}, x'_{i^*,j^*} - x_{i^*,j^*} \rangle \text{ for every } (i,j) \neq (i^*, j^*).$$

Moreover, since no other $x'_a - x_a$ with $a \neq a^*$ will appear in the image, we also have

$$\langle V^a, x'_{i,j} - x_{i,j} \rangle < 2p = \langle V^{a^*}, x'_{i^*,j^*} - x_{i^*,j^*} \rangle \text{ for every } i, j \text{ and action } a \neq a^*.$$

In other words, the max operator will pick the correct location for $a^*$ (first inequality) and the kernel response for $V^{a^*}$ at $(i^*, j^*)$ will be the largest (second inequality).

Let the score (pre-softmax) be defined as $z^a(s, s') := \max_{i,j} \langle K^a, (x_{i,j}, x'_{i,j}) \rangle$ for all $a$. Since $\langle K^a, (x_{i,j}, x'_{i,j}) \rangle = \langle V^a, x'_{i,j} - x_{i,j} \rangle$, the above inequalities imply that $z^{a^*}(s, s') > z^a(s, s')$ for all $a \neq a^*$. With $\hat{h}(a|s, s') = \text{softmax}(\tau^{-1} z(s, s'))$, we finally have that $\hat{h}(a|s, s') \to \mathbf{1}(a = a^*) = h^*(a|s, s')$ as $\tau \to 0$.

## D. Experiments

### D.1. Maze/grid Experiments

#### D.1.1. ARCHITECTURES

**LC - Linear Classifier (Figure 1 & 8, with $\mathcal{S}_{\text{pos}}$).** It corresponds to $\hat{\pi}(a \mid s) = \text{softmax}_a(Ws + b)$ where $W \in \mathbb{R}^{4 \times 2}$ and $b \in \mathbb{R}^4$ and $\hat{h}(a \mid s, s') = \text{softmax}_a(V(s, s') + c)$ were $V \in \mathbb{R}^{4 \times 4}$ and $b \in \mathbb{R}^4$.

**5L MLP - Multilayer Perceptron (Figures 1, 2, 3 & 8, with $\mathcal{S}_{\text{pos}}$).** It consists of a multilayer neural network with 5 hidden layers of 100 neurons with ReLU nonlinearities. The output is a softmax over the four possible actions. The input dimension depends on the setting: 2 inputs when used as a policy and 4 inputs when used as an IDM. In both cases, we add 2 input dimensions when goal conditioning (the goal is an $(x, y)$-location).

**1L CNN - One-layer convolutional neural network with global max pooling (Figure 1, with $\mathcal{S}_{\text{img}}$).** It has one convolutional layer with a 3x3 kernel, mapping the input channels (3 input channels for $\hat{\pi}(a \mid s)$ and 6 input channels for $\hat{h}(a \mid s, s')$) to 4 channels (one per action) without padding nor stride. Next, the maximum is taken across pixels (reducing the width and height dimensions), which results in a 4-dimensional vector which is then fed into a softmax function to obtain a distribution over actions.

**5L CNN - 5-layer convolutional neural network (Figure 1, with $\mathcal{S}_{\text{img}}$).** The number of input channels is 3 for $\hat{\pi}(a \mid s)$ and 6 for $\hat{h}(a \mid s, s')$. Three blocks of the form "convolution-ReLU-MaxPool" are applied, where the convolution has a 3x3 kernel with 128 output channels (padding=1, no stride) and the MaxPool layer has a 2x2 kernel (no padding, no stride). Then, two fully connected ReLU layers are applied, with 128 hidden units each. A final linear projection maps to a 4-dimensional vector, which is fed to a softmax to obtain a distribution over actions.

#### D.1.2. COMPLEXITY EXPERIMENTS (FIGURES 1 & 2) – ENVIRONMENTS, DATASETS & METRICS

The three mazes were randomly generated and solved using the `mazelab` library (Zuo, 2018). The action space is $\mathcal{A} := \{\texttt{right}, \texttt{left}, \texttt{up}, \texttt{down}\}$ and executing one of these deterministically moves the player in the corresponding direction (by one unit) unless a wall prevents this move, in which case the player remains static. Note that the resulting expert $\pi^*$ is always deterministic, allowing us to write $\pi^*(s)$ to denote the action taken by $\pi^*$ in state $s$. Moreover, the environment dynamics is also deterministic, i.e. $p(s' \mid s, a) = \mathbf{1}(s' = f(s, a))$ for some function $f$ where $\mathbf{1}(\cdot)$ is the indicator function, which means the ground-truth video model $v^*$ induced by the expert is also deterministic. Indeed,

$$v^*(s' \mid s) = \sum_a p(s' \mid s, a)\pi^*(a \mid s) = \sum_a \mathbf{1}(s' = f(s, a))\mathbf{1}(a = \pi^*(s)) = \mathbf{1}(s' = f(s, \pi^*(s))).$$

Define $v^*(s) := f(s, \pi^*(s))$.

With this notation setup, we have that the test set is given by

$$\mathcal{D}_L^{\text{test}} := \{(s, \pi^*(s), v^*(s)) \mid s \in \mathcal{S}_{\text{feasible}}\},$$

where $\mathcal{S}_{\text{feasible}}$ is the set of feasible states. The training set $\mathcal{D}_L^{\text{train}}$ is constructed by randomly sampling a fraction of the test set without replacement. In the goal conditioning experiment of Figure 2, the goal description $g$ is added as a feature.

The **test accuracy** of a learned policy $\hat{\pi}$ is given by

$$\text{Acc}_{\text{test}}(\hat{\pi}) := \frac{1}{|\mathcal{D}_L^{\text{test}}|} \sum_{(s,a,s') \in \mathcal{D}_L^{\text{test}}} \mathbf{1}(a = \hat{\pi}(s)),$$

where $\hat{\pi}(s) := \arg\max_a \hat{\pi}(a \mid s)$.

*Remark* D.1. In this deterministic setup, the IDM-based policy $\hat{\pi}_{v^*,\hat{h}}$ (Section 3) takes the particularly simple form $\hat{\pi}_{v^*,\hat{h}}(a \mid s) = \hat{h}(a \mid s, v^*(s))$. Indeed,

$$\hat{\pi}_{v^*,\hat{h}}(a \mid s) = \sum_{s'} \hat{h}(a \mid s, s') v^*(s' \mid s) = \sum_{s'} \hat{h}(a \mid s, s') \mathbf{1}(s' = v^*(s)) = \hat{h}(a \mid s, v^*(s)).$$

*Remark* D.2. The test accuracy of the IDM $\hat{h}$ is the same as the accuracy of the IDM-based policy $\hat{\pi}_{v^*,\hat{h}}$. Indeed, by defining $\hat{h}(s, s') = \arg\max_a \hat{h}(a \mid s, s')$, we get

$$\begin{aligned}
\text{Acc}_{\text{test}}(\hat{h}) &:= \frac{1}{|\mathcal{D}_L^{\text{test}}|} \sum_{(s,a,s') \in \mathcal{D}_L^{\text{test}}} \mathbf{1}(a = \hat{h}(s, s')) \\
&= \frac{1}{|\mathcal{D}_L^{\text{test}}|} \sum_{(s,a,s') \in \mathcal{D}_L^{\text{test}}} \mathbf{1}(a = \hat{h}(s, v^*(s))) \\
&= \frac{1}{|\mathcal{D}_L^{\text{test}}|} \sum_{(s,a,s') \in \mathcal{D}_L^{\text{test}}} \mathbf{1}(a = \hat{\pi}_{v^*,\hat{h}}(s)) = \text{Acc}_{\text{test}}(\hat{\pi}_{v^*,\hat{h}}).
\end{aligned}$$

*Remark* D.3. The definition and claims made above remain identical with goal conditioning, simply replace $s$ by $(s, g)$.

### D.1.3. STOCHASTIC EXPERT EXPERIMENT (FIGURE 3): ENVIRONMENTS, DATASETS & METRICS

As explained in Section 4.2, we consider a simple 20x20 grid environment where the agent starts at the top left corner and must reach the bottom right corner. The action space is again $\mathcal{A} := \{\texttt{right}, \texttt{left}, \texttt{up}, \texttt{down}\}$. There are no obstacles preventing movement, except for the borders of the grid. We consider three expert policies used to generate the training data. All three are optimal and consist in going right with probability $p(\texttt{right})$ and going down with probability $1 - p(\texttt{right})$, except when on the right or bottom border, where the policy deterministically chooses the only optimal action. We vary $p(\texttt{right})$ to study the effect of the stochasticity of the expert on the performance of BC and IDM labeling.

For each expert, we sample 26 full trajectories of length 38 (this is the number of actions required to solve the environment) for a total of 988 transitions $(s, a, s')$. These transitions form the dataset $\mathcal{D}^{\text{full}}$. The unlabeled dataset $\mathcal{D}_U$ is obtained by discarding the action labels in $\mathcal{D}^{\text{full}}$. The action-labeled training set $\mathcal{D}_L$ is formed by sampling a fraction of $\mathcal{D}_L^{\text{full}}$ without replacement ("train split" corresponds to that fraction). Note that the IDM $\hat{h}$ is trained on $\mathcal{D}_L$ while $\hat{\pi}$ is trained on $\mathcal{D}_U$ via the IDM labeling objective from Section 2.3.2.

The **average reward** measures the performance of a policy $\hat{\pi}$ by averaging the reward obtained over 25 episodes, each limited to 38 steps (the minimum necessary to reach the end goal). A reward of one is obtained if the goal is reached within 38 steps, otherwise the reward is 0.

*Remark* D.4. Note that accuracy is not a good metric to measure the performance of a policy $\hat{\pi}$ when the expert $\pi^*$ is stochastic. This is why we instead report the average reward.

### D.1.4. STOCHASTIC ENVIRONMENT EXPERIMENT (FIGURE 8)

In this set of experiments, we consider of variant of the 50x50 maze environment from Figure 1 where, at each time step, there is some probability $p(\texttt{no-op})$ that the agent will remain static, regardless of the action taken. The expert policy is the same as previously and is deterministic. This setting is interesting since the ground-truth IDM $h^*$ is not linear. Intuitively, this is because, when $s = s'$, the best thing $h^*$ can do is to do just like $\pi^*$, which is itself nonlinear. That being said, when

*Figure 8.* **Varying the stochasticity of the environment.** 50x50 maze from Figure 1 with deterministic expert and stochastic environment: the agent remains static with probability $p(\text{no-ops})$. Comparing the test accuracy of the of BC policy $\hat{\pi}_{\text{BC}}$ and the VM*-IDM policy $\hat{\pi}_{v^*,\hat{h}}$. Averaging over 5 seeds.

$s \neq s'$, the linear classifier described in Appendix C.1 works well. One can write the ground-truth IDM $h^*(a \mid s, s')$ as

$$\mathbf{1}(s = s')\pi^*(a \mid s) + \mathbf{1}(s \neq s')\text{softmax}_a(\tau^{-1}W(s' - s)),$$

which approaches $h^*(a \mid s, s')$ when $\tau \to 0$. It is clear that this function is not a linear classifier as soon as $\pi^*$ is not. Experiments will nevertheless show that this function is sufficiently close to being linear to be well modeled by a linear classifier.

Figure 8 shows again that the IDM-based policy is more sample efficient than the BC policy. Even if the ground-truth IDM is not linear, learning a linear IDM yields a very strong VM*-IDM policy, as long as the $p(\text{no-op}) < 1$. We hypothesize that this is due to the ground-truth IDM being linear on the region of the state space where $s \neq s'$, as explained above. The IDM implemented with an MLP is also able to leverage this near linearity, thanks to the simplicity bias of neural networks, which allows VM*-IDM to outperform BC. In the limit case where $p(\text{no-op}) = 1$, BC and VM*-IDM behave exactly the same since the linearity shortcut is never available to IDM since we always have $s = s'$.

**Accuracy computation.** We wish to compute the accuracy of the IDM-based policy $\hat{\pi}_{v^*,\hat{h}}$:

$$\text{Acc}(\hat{\pi}_{v^*,\hat{h}}) = \frac{1}{|\mathcal{D}_L^{\text{test}}|} \sum_{(s,a) \in \mathcal{D}_L^{\text{test}}} \mathbf{1}(a = \arg\max_{\hat{a}} \hat{\pi}_{v^*,\hat{h}}(\hat{a} \mid s)).$$

To achieve this, we first compute $\hat{\pi}_{v^*,\hat{h}}(a \mid s)$ explicitly using our knowledge of the environment. First, notice that

$$v^*(s' \mid s) = p(\text{no-op})\mathbf{1}(s' - s) + (1 - p(\text{no-op}))\mathbf{1}(s' - f(s, \pi^*(s))),$$

where $f(s, a)$ is the next-state in the case where the action is actually realized. From this, we can compute

$$\hat{\pi}_{v^*,\hat{h}}(a \mid s) = \sum_{s'} \hat{h}(a \mid s, s')v^*(s' \mid s)$$

$$= \sum_{s'} \hat{h}(a \mid s, s')[p(\text{no-op})\mathbf{1}(s' - s) + (1 - p(\text{no-op}))\mathbf{1}(s' - f(s, \pi^*(s)))]$$

$$= p(\text{no-op})\sum_{s'} \hat{h}(a \mid s, s')\mathbf{1}(s' - s) + (1 - p(\text{no-op}))\sum_{s'} \hat{h}(a \mid s, s')\mathbf{1}(s' - f(s, \pi^*(s)))$$

$$= p(\text{no-op})\hat{h}(a \mid s, s) + (1 - p(\text{no-op}))\hat{h}(a \mid s, f(s, \pi^*(s))).$$

The accuracy is computed using this expression for $\hat{\pi}_{v^*,\hat{h}}(a \mid s)$.

### D.1.5. OPTIMIZATION

In all maze/grid experiments (Figures 1, 2 & 3), we train using the Adam optimizer. To train LC and 5L MLP, we use a learning rate of 1e-3, while we use 1e-4 for 1L CNN and 5L CNN. For the CNN architecture we use a batch size of $\min\{32, |\mathcal{D}_L|\}$ while we use a batch size of $|\mathcal{D}_L|$ for the MLP experiments, except for the policies of Figure 3, where we used a batch size of $\min\{512, |\mathcal{D}_L|\}$ for BC and $\min\{512, |\mathcal{D}_U|\}$ for IDM Labeling.

### D.2. ProcGen Experiments

#### D.2.1. TRAINING DETAILS

In ProcGen environments, $s$ and $s'$ are RGB frames and $a$ is a discrete action. We use the data and learning setup from LAPO (Schmidt & Jiang, 2024), relying on IMPALA-CNNs (Espeholt et al., 2018) followed by a fully-connected action

decoder with hidden sizes (128, 128) to implement BC, IDM labeling, LAPO and LAPO+. We still train the action decoder (end-to-end with the backbone) during BC and IDM labeling for parity. We use a learning rate of 2e-4 (except for LAPO/LAPO+ stage 1 where we use 3e-4), a piecewise linear schedule, a batch size of $\min\{128, |\mathcal{D}_L|\}$ (or 128 if only $\mathcal{D}_U$ is used) and the Adam optimizer. To ensure a fair comparison, we make sure the total number of training steps across all stages sums to 120,000 for all methods. Specifically, we use 120,000 steps for BC, 60,000/60,000 steps for the two stages of IDM labeling, 50,000/60,000/10,000 steps for the three LAPO stages, and 50,000/10,000/60,000 steps for the three LAPO+ stages. During policy learning stages, we rollout the policy every 500 steps and average the return over 64 episodes, keeping the maximum episodic return achieved during training for Figure 5.

For latent action policies, we do not retrain LAPO stage 1 for each seed and instead use the same stage 1 models (one for each environment) for all downstream LAPO and LAPO+ runs. We reuse the quantization hyperparameters from (Schmidt & Jiang, 2024) for LAPO stage 1. While LAPO originally included one additional frame of pre-transition context for the LIDM, we do not include any pre-transition context to simplify the comparison between IDM-based methods and BC.

### D.2.2. COMPLEXITY CLASSIFICATION

We consider the `bigfish`, `bossfight`, `dodgeball`, `fruitbot`, `heist`, `maze`, `miner`, `plunder` and `starpilot` environments as having simple dynamics. We consider the following environments as complex: `caveflyer` and `chaser` because of momentum (agent does not stop immediately given no inputs), `climber`, `coinrun`, `jumper` and `ninja` because of gravity, and `leaper` because of moving platforms.

### D.3. Manipulation Experiments

For Push-T and LIBERO-10, $s$ and $s'$ include four RGB frames each and $a$ is a block of 16 continuous action vectors. We use the UVA codebase (Li et al., 2025). UVA is initialized with the pretrained image generation model MAR-B (Li et al., 2024) and encodes all images using a frozen VAE KL-16 (Rombach et al., 2022). The architecture relies on a transformer backbone and two separate diffusion heads for video generation and action generation. We refer to the original paper for details. The UVA architecture implements the following task modes through input masking and selective output decoding: policy, VM, FDM, IDM and a *full dynamics model* (i.e., $p(a, s'|s)$). We modified the UVA masking procedure to include all current frames $s$ in the IDM to put it on par with the policy. For a given batch during training, UVA uniformly samples a task, applies masking to the irrelevant inputs, and computes the loss for the sampled task. We use a learning rate of 2e-5, a batch size of 128, and train all models for 50,000 steps using the AdamW optimizer on nodes with 4xH100 GPUs.

We pretrain UVA in VM mode only on $\mathcal{D}_U$ for each environment and use this as an initialization for all methods, per the UVA training recommendations. As such, all methods in this section perform SSIL via their initialization. We then disable the other task modes to learn single functions and perform BC or IDM learning on $\mathcal{D}_L$. Next, we use the same IDMs for IDM labeling (again starting from the VM initialization and fitting UVA in policy mode on the IDM-labeled $\mathcal{D}_U$) or VM-IDM (pairing the IDMs with the initial VM). We also train standard UVA with multitasking on $\mathcal{D}_L$ and show the performance of the two sampling paths Policy (UVA) and Video-IDM (UVA).

We experienced relatively high variance when training with diffusion heads on LIBERO-10 (perhaps due to the limited size of $\mathcal{D}_L$) and chose to include results of a UVA variant with a linear action head due to its overall stronger performance on this benchmark. We use a `SmoothL1` loss with $\beta = 1.0$ in this case.

