# OpenReview forum: "On the Sample Efficiency of Inverse Dynamics Models for Semi-Supervised Imitation Learning"
_ICML.cc/2026/Conference — ICML 2026 regular_

### Official Review · Reviewer_hfYo · 2026-03-03

**Soundness:** 2
**Presentation:** 3
**Significance:** 2
**Originality:** 2
**Overall Recommendation:** 4
**Confidence:** 3

**Summary:**

The paper investigates semi-supervised imitation learning, especially two types of algorithm: VM-IDM and IDM labeling. The paper demonstrates their connection under a limit case that they both learn the same IDM-based policy. The paper then further investigates the superiority of IDM-based policy compared to BC policy by providing two claims: (1) ground-truth IDM tends to have lower hypothesis class relative to expert policy. (2) the ground-truth IDM is often less stochastic than expert policy. Based on the two claims, the paper provides an improvement based on LAPO algorithm and conduct empirical evaluations on multiple robotic manipulation benchmarks.

**Compliance With Llm Reviewing Policy:**

Affirmed.

**Final Justification:**

Much of the concerns are addressed in the rebuttal period.

**Key Questions For Authors:**

See the weakness section.

**Limitations:**

yes

**Strengths And Weaknesses:**

## Strengths

- The paper provides extensive reasoning and empirical validations on their claims on why IDM-based methods are better than BC. The authors support the reasoning by some analysis based on toy constructions.

- The paper is overall well-written and easy to follow. Some visualizations on the experimental results support the claims.

- The improved version of LAPO does have some improvement over the original version in most tasks.

## Weaknesses

- My key concern lies on the theory aspect of this paper.
1. Some theoretical justifications, especially for those regarding the sampling complexity comparison, are not satisfying to me. For example. in Eq. 5, this is not a standard way of comparing sampling complexities in learning theory. Two statistical bounds: $D_{KL}(h^*\Vert h)=O(|H|/n^p)$ and $D_{KL}(\pi^*\Vert \pi)=O(|\Pi|/n^q)$ should be derived, and given an $\epsilon$-optimal policy, the order of $n$ can be compared. The authors should consider changing the sampling complexity analysis to this standard form.

2. The statement of generalization via the simplicity bias of NN in Page 5 is vague. More theoretical justification on SGD implicit bias within this setting should be derived.

3. The authors should consider providing a more general analysis aside from a toy case construction in "simple analytic form for $h^*$ but not for $\pi^*$.

- Regarding the empirical results of LAPO+, in some settings (e.g. leaper and plunder), LAPO+ has lower score than BC or IDM labeling on  large data regime. The author states that it's potentially associated with latent action policies. Could the authors discuss in details why latent action policy would suffer from this scenario?

---

> ### Author Rebuttal · Authors · 2026-03-30
>
> We thank the reviewer for engaging with our work and providing valuable feedback.
>
> **Sample complexity comparison (Equation 5).**
>
> The reviewer makes the interesting observation that our sample complexity definition is not standard and suggests deriving alternative bounds.
>
> Equation 5 currently bounds the estimator/ground-truth KL divergence of the IDM with the estimator/ground-truth KL divergence of the BC policy for a particular $N_L$ (via the estimators). We stress that Equation 5 is not a result and rather a definition which aims to formalize of the statement "IDM learning is more sample efficient than BC" made in previous works, which emphasizes the **relative sample complexity**. To make the sample complexity aspect more explicit, we updated the paper to clarify that Equation 5 holds for most dataset sizes $N_L$ with high probability. We will also add ${}^{(N_L)}$ superscripts in Equation 5 for clarity. Our definition is **general** and critically allows to characterize the sample complexity of the induced IDM-based policy $\hat{\pi}\_{v^\*, \hat{h}}$ compared to $\hat{\pi}_{BC}$ (lines 180-189).
>
> The bounds proposed by the reviewer are more specific and would allow to compare convergence rates, which is certainly interesting. However, we consider this to be out of scope, as the novelty of our contribution lies in using the knowledge from statistical learning theory to explain the good performance of popular IDM-based methods in SSIL, as opposed to proving formal sample complexity results.
>
> **Statement of generalization via the simplicity bias.**
>
> The reviewer considers that our simplicity bias statement is too vague and requests additional theoretical justification that are specific to the IDM learning setting. While such developments would be valuable, we believe they are not strictly necessary to make our case since we can refer to the now vast literature on the topic. We recall our argument: in the maze examples, we show that the ground-truth IDMs are simple functions (linear when states are positions, 1-layer CNN when states are images) and empirically find they can be learned with few samples using overparametrized models (5-layer MLP, 5-layer CNN). For policy learning, it is clear that the expert $\pi^\*$ does not admit such a simple form and we empirically find that these "high-capacity" architectures require much more data. These findings are well explained by the simplicity bias of neural networks since sample efficiency correlates with simplicity of the target function. Our argument relies on existing investigations, both theoretical and empirical, which proposed mechanisms (SGD bias, parameter-to-function map) for this simplicity bias. It is likely that many such mechanisms are in action in our setting.
>
> The reviewer's observation also sparked interesting discussions on alternative explanations for the simplicity bias in the context of IDM learning. One such explanation has been proposed in [1], which shows empirically that SGD visits linear classifiers early during training, and more generally learns functions of increasing complexity that retain information from the previous ones. This explanation is particularly interesting in the context of a linear ground truth IDM $h^\*$ and non-linear ground truth policy $\pi^\*$.
>
> **More general analysis than the simple analytic form.**
>
> Our simple analytic forms serve as motivating examples and guide the controlled experiments. Our position-based maze example is a particular example of a more general setting where $h^\*$ is a linear function and $\pi^\*$ is not. We extend this to the **image-based case** (Appendix B.2) which, while still arguably simple, is already less toy. In the image case, the ground-truth IDM $h^\*$ can be represented analytically as a single layer CNN while it appears the policy $\pi^\*$ cannot (as Fig 1 suggests).
>
> A more general analysis would be challenging: IDMs can vary substantially depending on the environment and the observation space, and they can be hard to write down analytically (which is why we learn them, after all). Our maze examples still serve as useful illustrations capturing the essence of our argument. We study a wider range of environments empirically in Section 5.
>
> **LAPO+ and LAPO performance in large data regimes.**
>
> Both LAPO+ and LAPO learn a latent IDM during stage 1 on action-free data, learning latent actions/representations for the transitions $(s, s')$. The IDM is then frozen for the rest of the algorithms and therefore never benefits from the increased availability of labeled data. In particular, the learned frozen representations prove valuable in low-data regimes, but potentially does not include enough information to perfectly decode actions and be competitive with IDM labeling and BC in high-data regimes. See our answer to reviewer J5ML for a detailed discussion.
>
> [1] Nakkiran, Preetum, et al. "SGD on neural networks learns functions of increasing complexity." arXiv:1905.11604 (2019).

---

> > ### Author Rebuttal · Reviewer_hfYo · 2026-04-03
> >
> > My concerns have been adequately addressed.

---

### Official Review · Reviewer_W9Cz · 2026-03-12

**Soundness:** 2
**Presentation:** 3
**Significance:** 3
**Originality:** 2
**Overall Recommendation:** 4
**Confidence:** 3

**Summary:**

The authors study semi-supervised imitation learning, where a small dataset of action-labeled
transitions and a large dataset of action-free state transitions are available. The manuscript analyzes
two approaches that use inverse dynamics models (IDMs) to infer actions from state transitions and
shows that under idealized conditions, these approaches learn the same policy ("IDM-based policy").
Furthermore, the authors present a few arguments why IDMs could be more sample-efficient than
behavioral cloning, mainly around the complexity of learning policies and inverse dynamics, and
validate them in a simplified gridworld maze environment. Finally, the authors perform an empirical
analysis in more complex ProcGen and manipulation environments and interpret the observed
results through the discussed notions of environment and policy complexity.

**Compliance With Llm Reviewing Policy:**

Affirmed.

**Final Justification:**

authors gave good clarifications, adjusted score

**Key Questions For Authors:**

1. To the best of my understanding the analysis in section 3 is valid. However, the nuance of
this analysis is that it considers asymptotic behavior and does not describe the sample
efficiency in which the IDM policies are learned. Do I have a correct understanding?

2. Equation 5 presents a sensible bound on the performance of the policies achieved by BC
and IDM, but I do not see how it is supposed to describe sample efficiency. If we want to
consider sample efficiency, shouldn't we study policies different from $\pi^{\star}$?

3. The arguments and analysis presented in Section 4.1.1 is valuable and interesting. However,
it is worth noting that the linear separability in the inverse dynamics is not a given, and is
especially visible in a deterministic maze with four actions. At the same time, authors often
use general statements like "This suggests IDM learning is more sample-efficient than BC".
Authors should be more upfront about the limitations of this analysis and the relationship
between the environment design and conclusions. Or if the authors want to make their
argument more general, why not run this analysis on OGBench (eg humanoid-maze where
the dynamics could include humanoid control) or something more complex? Or is using
statements like "This suggests IDM learning is more sample-efficient than BC" justified in the
author's opinion?

4. To the best of my understanding the analysis assumes that the expert policy is stochastic,
whereas the environment is deterministic. We can imagine a reverse case with deterministic
policy and stochastic dynamics. Would the analysis presented in Figure 3 hold?

**Limitations:**

yes

**Strengths And Weaknesses:**

Strengths:
1. The key ideas are delivered in a clear way
2. The experimental analysis does not present any cutting edge results, but serves a good role
in strengthening the analysis
Weaknesses:
1. Some conclusions are presented as general for IDMs, while the supporting analysis is based
on environments with simple, deterministic inverse dynamics.
2. Since the authors base their arguments mainly of empirical observations, I think that the
experimental analysis should be broader to make these arguments more convincing

---

> ### Author Rebuttal · Authors · 2026-03-30
>
> We sincerely thank the reviewer for their important comments. They triggered interesting discussions on our side which we believe ultimately improved our understanding. See **new experiments** below.
>
> **Limitations + Relationship between the environment design and conclusions**
>
> We’re happy to read that the reviewer thought “The arguments and analysis presented in Section 4.1.1 is valuable and interesting” and we agree that it is not always true that “IDM learning is more sample-efficient than policy learning”. In fact, one of the main goals of our work is to explore precisely “**when** and to what extent IDM-based learning can outperform BC” (cited from page 1 bottom right). We make this goal explicit also at line 192, left column: “While the superior sample efficiency of IDM learning has been observed empirically in Minecraft and in the context of VM-IDM policies for robotics, it is not entirely clear **when** and why this should be expected.” We thus do not claim IDM learning is always more sample-efficient, but aim to understand when and why it will be. This requires full control over the environment and expert policy, making simple maze settings a natural candidate. More realistic environments rarely afford this level of control, making it harder to isolate factors like environment complexity, goal conditioning, and policy stochasticity. We did study some of these factors in the more challenging ProcGen setting (Fig 5).
>
> That being said, we plan on expanding the discussion on limitations to include the point raised above, to clarify that our simple setup with linear IDMs acts as a metaphor to think about more complex problems and to recognize that controlled experiments with more realistic environments would be needed to make our arguments more airtight. Given this limitation, we still believe our work improves our understanding of popular IDM-based methods for SSIL in a meaningful way.
>
> Also, we made many small modifications to avoid making too general claims. For example, on line 234, we change to “This suggests IDM learning is more sample-efficient than BC **in this setting**”.
>
> **Impact of environment stochasticity (NEW EXPERIMENTS)**
>
> We thank the reviewer for this interesting suggestion. We address this using the lens proposed in our work: How does the stochasticity of the environment impact the relative complexity/stochasticity of $h^\*$ and $\pi^\*$? When the expert policy is deterministic (as suggested by the reviewer), the ground-truth IDM must also be deterministic, since $0 \leq H(a | s, s’) \leq H(a | s) = 0$. So stochasticity doesn’t intervene here. What about complexity? We give two different cases of environmental stochasticity in our 50x50 maze of Fig 1.
>
> **Case 1:** Each action moves the agent in the desired direction by 1 or 2 steps with equal probability. Here, the $s' − s$ trick still applies, so $h^\*$ remains linear and we find experimental results identical to Fig 1 (top right).
>
> **Case 2:** Actions are not executed with some probability (no-ops). Here, the $s' − s$ linear trick does not hold. Experiment results are at https://postimg.cc/t1fjDCfQ. When p(no-op) = 1, BC and VM-IDM behave identically since $s=s'$ always. For 0<p(no-op)<1: (1) VM-IDM with linear IDM outperforms other methods at small data sizes (since the linear heuristic based on $s'−s$ performs decently) but is overtaken at large sizes since more capacity is needed to get perfect accuracy. (2) VM-IDM (MLP) outperforms BC (MLP) at small dataset sizes but is slightly overtaken at large ones. We attribute this to two opposing forces: (i) the MLP IDM quickly exploits the linear heuristic (thanks to NN simplicity bias), which BC cannot directly implement; (ii) the IDM must learn to ignore the noisy feature $s'$ to achieve perfect accuracy, requiring more samples than BC.
>
> **Equation 5 & $\pi^\*$**
>
> In Equation 5, the Kullback-Leibler divergences measure the distances between the learned policy/IDM (learned with finite data) and their ground-truth counterparts ($\pi^\*$ / $h^\*$). The inequality itself states that the learned IDM is closer to the ground-truth IDM than the BC policy is to the ground-truth expert policy. We have added a superscript $^{(N_L)}$ next to the learned policy/IDM to emphasize that they depend on the number of action-labeled transitions (finite data).
>
> **Analysis in Section 3**
>
> Indeed, this section does not describe the sample efficiency of IDM learning v.s. policy learning. We consider the limit case of infinite *unlabeled* data throughout this section, but we do not assume we have an infinite amount of *action-labeled* data. In fact, we define the IDM-based policy as $\hat{\pi}_{v^\*, \hat{h}}$ where $v^\*$ refers to the ground-truth video model (infinite unlabeled data) and $\hat{h}$ refers to the IDM learned with *finite* action-labeled data. The very last paragraph of the section considers the case where an infinite amount of labeled data is available and relates to the expert policy.

---

> > ### Author Rebuttal · Reviewer_W9Cz · 2026-04-03
> >
> > good clarifications

---

### Official Review · Reviewer_FAP1 · 2026-03-13

**Soundness:** 2
**Presentation:** 2
**Significance:** 2
**Originality:** 3
**Overall Recommendation:** 3
**Confidence:** 3

**Summary:**

This article discusses the theme of sample efficiency of inverse dynamics models (IDMs) in semi-supervised imitation learning. It unifies VM-IDM and IDM-labeling as recovering the same “IDM-based policy” in the infinite-unlabeled-data limit, then uses statistical learning theory to explain why IDMs generalize better than behavior cloning. Experiments present the effectiveness of the proposed method through controlled maze, ProcGen suite, Push-T and Libero10.

**Compliance With Llm Reviewing Policy:**

Affirmed.

**Final Justification:**

According to the rebuttal from the authors, I understand that UVA is treated as a variant of IDM in this paper, rather than the core contribution. However, the UVA-related experiments constitute roughly half of the main experimental section. In this context, the lack of vision-based benchmarks (e.g., robomimic, ManiSkill) remains a notable drawback. In my view, quantitative results on such benchmarks would be necessary to fully support the claims, and the current results are not sufficient to compensate for this gap. That said, considering the other reviewers’ assessments and recognizing that the UVA part does not fundamentally affect the main proposed algorithm pipeline, I am willing to adjust my recommendation from Reject to Weak Reject.

**Key Questions For Authors:**

1.Given the emphasis on video models in the paper, why were additional standard vision-based robotics benchmarks (e.g., robomimic, ROBOTWIN, ManiSkill) omitted? Results on any of these would substantially increase confidence in the generalizability of the VM-IDM claims. If the authors can provide even preliminary numbers, my assessment of the paper's empirical soundness would improve markedly.

2.The proposed modifications to LAPO and UVA outperform their respective baselines, which makes the effectiveness of the approach more convincing. However, if the method can be combined with latent-action approaches in principle, additional experiments on other latent-action methods would help demonstrate its broader applicability and strengthen the claims of generalizability.

**Limitations:**

yes

**Strengths And Weaknesses:**

Strengths：
1. The theoretical unification of VM-IDM and IDM labeling into the IDM-based policy, together with the clean KL-divergence arguments and the inequality linking IDM error to policy error, is technically solid and gives a principled explanation for a phenomenon that was previously only observed empirically.
2. The empirical investigation on controlled maze experiments isolate complexity and stochasticity effects, ProcGen provides a broad discrete-action stress test, and manipulation results with modern UVA architectures show the insights transfer to continuous control and diffusion policies.

Weaknesses：
1. While the paper repeatedly highlights video models (VM) and unified video-action architectures, the vision-based benchmark experiments are limited to Libero10 only; popular vision benchmarks such as robomimic, robotwin, and maniskill are completely not done. This is a notable gap for a work that positions itself around leveraging abundant action-free video data.

2. Some results (especially diffusion-head variants on Libero10) show high seed-to-seed variance, and the paper does not always quantify how sensitive the reported gains are to hyper-parameters or architecture details.

---

> ### Author Rebuttal · Authors · 2026-03-30
>
> We thank the reviewer for the feedback. The reviewer wrote a generally positive review, highlighting both the **theoretical unification** and the **empirical investigation** presented in our work.  We address key questions and weaknesses below. We also provide **new experiments on a robomimic task.**
>
> **More vision-based benchmarks. (NEW EXPERIMENTS)**
>
> We first want to clarify that the Push-T results, while in a simpler environment, are also vision-based.
>
> The reviewer suggests relevant vision-based robot learning benchmarks. While IDM learning is particularly relevant to robotics, our primary goal was to study the performance of IDM-based methods in different settings, as opposed to running a full-fledged robot learning evaluation. Our experimental results already cover toy maze environments, 16 Procgen environments, Push-T, and Libero-10 (a goal-conditioned benchmark with 10 distinct goals), for a total of 28 tasks.
>
> Training UVA models on new tasks is particularly expensive due to the video pretraining stage, which is required to initialize all other methods (including BC and the original UVA). Nevertheless, we managed to obtain preliminary results on the **Tool Hang task from robomimic**.
>
> Using 100 labeled demonstrations (approximately 50\% of the available data), we observe a success rate of 52\% for BC and 56\% for VM-IDM for a single seed. Interestingly, we observe a much lower performance of 42\% for UVA (Policy) and 30\% for UVA (VM-IDM) on this task. While the VM-IDM results are encouraging, more seeds and analysis would be required to conclude. We hope to study IDM-based policies more rigorously on similar long-horizon manipulation tasks in the future.
>
>
> **Other latent action methods.**
>
> The reviewer correctly points out that the LAPO+ modifications could potentially be extended to other latent-action methods and suggests experimenting with said methods.
> We focused on improving the offline variant of LAPO specifically and replicated their evaluation on ProcGen over a wider range of $\mathcal{D}_L$ sizes, finding that LAPO+ consistently outperforms LAPO. We note that LAPO-like algorithms have proven successful in settings such as real-world robotics [1], and agree that testing LAPO+ on similar problems would broaden the applicability of our method, but leave this to future work.
>
> **High seed-to-seed variance.**
>
> We subsample a small labeled dataset $\mathcal{D}_L$ for each seed which explains some of the variance in the final scores. Moreover, we generally experienced high success rate variance when training the UVA transformer architecture for all considered methods (BC, IDM-based policies, and UVA). We observe this variance despite using the default hyperparameters from the UVA codebase, with the exception of the learning rate, which we reduced to 2e-5 to improve stability. While studying and reducing the variance of UVA training is of great interest, it is not currently a core focus of our work.
>
> [1] Ye, Seonghyeon, et al. "Latent action pretraining from videos." arXiv preprint arXiv:2410.11758 (2024).

---

> > ### Author Rebuttal · Reviewer_FAP1 · 2026-04-07
> >
> > Thank the authors for their response. I understand that UVA is treated as a variant of IDM in this paper, rather than the core contribution. However, the UVA-related experiments constitute roughly half of the main experimental section. In this context, the lack of vision-based benchmarks (e.g., robomimic, ManiSkill) remains a notable drawback. In my view, quantitative results on such benchmarks would be necessary to fully support the claims, and the current results are not sufficient to compensate for this gap.
> > That said, considering the other reviewers’ assessments and recognizing that the UVA part does not fundamentally affect the main proposed algorithm pipeline, I am willing to adjust my recommendation from Reject to Weak Reject.

---

### Official Review · Reviewer_J5ML · 2026-03-13

**Soundness:** 3
**Presentation:** 3
**Significance:** 4
**Originality:** 3
**Overall Recommendation:** 5
**Confidence:** 5

**Summary:**

The paper studies inverse-dynamics-model-based semi-supervised imitation learning and asks when IDM-based methods can be more label-efficient than standard behavior cloning. The main claim is that IDMs can enjoy a statistical advantage because the ground-truth inverse dynamics is often both simpler and less stochastic than the expert policy, and the paper formalizes this view by relating VM-IDM and IDM labeling. The authors then use the resulting insights to motivate an improved latent-action method, LAPO+, evaluated on ProcGen, Push-T, and Libero.

**Compliance With Llm Reviewing Policy:**

Affirmed.

**Final Justification:**

My opinion is that this work can bring value to the community. I will therefore maintain my score.

**Key Questions For Authors:**

Questions:
- Could the authors clarify what the ground truth video model v^star is in the VM-IDM setup, especially since it is used to produce the plots in Fig 1?
- L377-378 mention "LAPO and LAPO+ tend to underperform in high-data regimes". Could the authors elaborate on why this may happen? In particular, why do they view this phenomenon as somewhat orthogonal to their main claim that IDM-based learning benefits from lower complexity and lower stochasticity?

**Limitations:**

yes

**Strengths And Weaknesses:**

Strengths:
- The paper addresses a timely problem with potentially high practical impact: scaling behavior cloning is bottlenecked by the need for action-labeled expert demonstrations.
- The empirical evaluation covers a reasonably broad range of tasks, including both discrete and continuous control settings.
- The introduction is clear, and the paper is generally easy to follow.
- The integration of related work and background is effective and works well for this paper.


Weaknesses:
- I did not find the bias-variance tradeoff discussion following Claim 1 very convincing or sufficiently substantive in supporting the claim.
- The "Generalization via capacity control" block is fairly standard and generic; moving it to the appendix could improve the focus of the main text.
- L304-305 mention "VM-IDM without goal-conditioning", but in Fig 2 the VM-based methods appear goal-conditioned according to the legend. This seems inconsistent and should be clarified.


Minor:
- (Maybe) L57: missing ":".

---

> ### Author Rebuttal · Authors · 2026-03-30
>
> We warmly thank the reviewer for engaging with our work and supporting its acceptance to ICML 2026. Below, we address the points raised in their review.
>
> **What's the ground truth video model $v^\*$ in the VM-IDM setup from Fig 1?**
>
> Since both the expert policy and the environment are fully deterministic, the ground-truth video model $v^\*$ is also fully deterministic. There was no need to train a model explicitly to map from $s$ to $s'$, since we always have that $s' = v^\*(s)$ and thus the transitions $(s, s')$ from the dataset are exactly equal to $(s, v^\*(s))$. So we can simply take $s'$ from the observed transition to be the output of the deterministic video model $v^\*(s)$.  We explain this in much more details in Appendix C.1.2. See in particular Remark C.2 which establishes that the accuracy of the IDM-based policy $\hat\pi_{v^\*, \hat h}$ is equal to the accuracy of the learned IDM $\hat h$ in this fully deterministic setting.
>
> **L304-305: VM is goal-conditioned**
>
> We agree that “VM-IDM without goal-conditioning” is misleading since the VM is always goal-conditioned. We rephrased this paragraph to avoid confusion:
> “[...] We explore the impact of goal-conditioning for the BC and VM-IDM policies. We look at the BC policy with and without goal-conditioning, that is with and without $g$ as input. We also look at the VM-IDM policy where the IDM is either goal-conditioned or not and the VM is always goal-conditioned and equal to the ground-truth [...]”
>
> **The "Generalization via capacity control" block is generic, should move to the appendix**
>
> We agree this section is fairly standard. That being said, we are hoping to make the paper readable and pedagocial to people who might not have a good understanding of these notions. We also believe stating these argument explicitly is important given how central this point is to our narrative.
>
> **LAPO and LAPO+ underperform in high-data regimes**
>
> Thanks for this question! Our hypothesis for this observation is the following: The stage 1 of LAPO and LAPO+ (which are identical) learns a latent IDM which extracts a latent action/representation from the pair $(s, s’)$. In LAPO+, a decoding head is learned on top of this latent IDM to map to actual actions, where the latent IDM is frozen. A priori, there’s no reason to believe that there exists a decoding head that allows the decoded latent IDM to express exactly the ground-truth IDM, which would mean no amount of action-labeled data would make the decoded IDM approach the ground-truth IDM, hence the gap in performance (since bad IDM means bad IDM-based policy). One reason why this problem might occur is if, e.g., the latent IDM does not extract enough information from the pair $(s, s’)$ so that the actual action cannot be predicted only from the latent representation outputted by the latent IDM. The situation is analogous in original LAPO, where an imperfect latent IDM would translate in an imperfect latent policy, for which no decoder might be able to represent the expert policy properly. We will add a small discussion addressing this in the next version of our manuscript.
>
> **Claim 1**
>
> We appreciate the reviewer's feedback on the bias-variance tradeoff discussion following Claim 1. We are interested in strengthening this section and nuancing the claim if necessary. To do so effectively, it would be helpful to know which specific parts of the argument the reviewer found less convincing.

---

> > ### Author Rebuttal · Reviewer_J5ML · 2026-04-03
> >
> > The rebuttal addressed my questions satisfactorily and clarified the points I had flagged, including the goal-conditioning issue and the interpretation of the VM-IDM setup.
> >
> > The authors also committed to revising the presentation to address the clarity concerns.
> >
> > After re-reading the discussion following Claim 1, I do not have a fundamental concern with the argument; my remaining reservation is mostly about exposition, which could be made lighter and more direct.
> >
> > The response on the high-data behavior of LAPO/LAPO+ is plausible.
> >
> > Overall, I continue to view the paper as technically solid, and I believe the analysis is conducted carefully.

---

### Decision · Program_Chairs · 2026-04-30

**Decision:**

Accept (regular)

**Comment:**

This paper studies when inverse dynamics models provide higher sample efficiency than behaviour cloning. This is an important question, especially given recent work showing that variants of the VM‑IDM approach studied here achieve state‑of‑the‑art performance on challenging robotics benchmarks.

During the review process, the results were generally seen as sound and clearly presented, and reviewers viewed the paper as helpful for practitioners seeking to understand when inverse dynamics is likely to be beneficial. At the same time, some concerns were raised. A common weakness was the limited generality of the conclusions, which depend on specific tasks, assumptions, and experimental settings. During the rebuttal, the authors clarified their claims, addressed specific experimental questions, and emphasised that the paper is intended as a theoretically-motivated empirical study to better understand practice rather than to propose a new algorithm. Most of the original concerns were addressed satisfactorily.

One remaining issue was the lack of experiments in vision‑based settings, which would help further support the paper’s narrative. However, this is not critical, as recent studies have already demonstrated strong performance of the architecture under consideration on challenging robotics tasks with multi‑camera inputs.

Overall, while the paper does not fully resolve questions of theory or broad generality, it provides useful empirical evidence and insights into sample efficiency in imitation learning, and is likely to be of interest to the reinforcement learning and robotics communities.